# Quantitative proteomic analysis of skeletal muscles from wild-type and transgenic mice carrying recessive *Ryr1* mutations linked to congenital myopathies

Jan Eckhardt[1†‡], Alexis Ruiz[1†], Stéphane Koenig[2], Maud Frieden[2], Hervé Meier[1], Alexander Schmidt[3], Susan Treves[1,4]*, Francesco Zorzato[1,4]*

[1]Departments of Biomedicine and Neurology, Basel University Hospital, Basel, Switzerland; [2]Department of Cell Physiology and Metabolism, University of Geneva, Geneva, Switzerland; [3]Proteomics Core Facility, Biozentrum, Basel University, Basel, Switzerland; [4]Department of Life Science and Biotechnology, University of Ferrara, Ferrara, Italy

*For correspondence:
susan.treves@unibas.ch (ST);
fzorzato@usb.ch (FZ)

†These authors contributed equally to this work

Present address: ‡Friedrich Miescher Institute for Biomedical Research, Maulbeerstrasse, Basel, Switzerland

Competing interest: The authors declare that no competing interests exist.

**Abstract** Skeletal muscles are a highly structured tissue responsible for movement and metabolic regulation, which can be broadly subdivided into fast and slow twitch muscles with each type expressing common as well as specific sets of proteins. Congenital myopathies are a group of muscle diseases leading to a weak muscle phenotype caused by mutations in a number of genes including *RYR1*. Patients carrying recessive *RYR1* mutations usually present from birth and are generally more severely affected, showing preferential involvement of fast twitch muscles as well as extraocular and facial muscles. In order to gain more insight into the pathophysiology of recessive *RYR1*-congential myopathies, we performed relative and absolute quantitative proteomic analysis of skeletal muscles from wild-type and transgenic mice carrying p.Q1970fsX16 and p.A4329D RyR1 mutations which were identified in a child with a severe congenital myopathy. Our in-depth proteomic analysis shows that recessive *RYR1* mutations not only decrease the content of RyR1 protein in muscle, but change the expression of 1130, 753, and 967 proteins EDL, soleus and extraocular muscles, respectively. Specifically, recessive *RYR1* mutations affect the expression level of proteins involved in calcium signaling, extracellular matrix, metabolism and ER protein quality control. This study also reveals the stoichiometry of major proteins involved in excitation contraction coupling and identifies novel potential pharmacological targets to treat RyR1-related congenital myopathies.

## Editor's evaluation

This is a fundamental study reporting a comprehensive proteomic analysis in three skeletal muscle types from wild-type and RYR1-related myopathy mice. It adds quantitative stoichiometry of several excitation-contraction coupling-related proteins. This valuable work compares the disease-related proteomes of the different skeletal muscle groups.

## Introduction

Skeletal muscles constitute the largest organ, accounting for approximately 60% of the total body mass; they are responsible for movement and posture and additionally, play a fundamental role in regulating metabolism. Furthermore, skeletal muscles are plastic and can respond to physiological

stimuli such as increased workload and exercise by undergoing hypertrophy. Broadly speaking muscles can be subdivided into different types depending on their speed of contraction, namely slow twitch muscles are characterized by level of oxidative activity, while fast twitch muscles show high content of enzymes involved in glycolytic activity. Fast- and slow-twitch muscle can be also identified based on the expression of specific myosin heavy chain (MyHC) isoforms (*Lieber, 2010*; *Schiaffino and Reggiani, 2011*). Fast twitch muscles, also known as type II fibers, are specialized for rapid movements, are mainly glycolytic contain large glycogen stores and few mitochondria, fatigue rapidly and characteristically express the MyHC isoforms 2 X, 2B, and 2 A. They are also the first muscles to appear during development and are more severely impacted in patients with congenital myopathies; they also undergo more prominent age-related atrophy or sarcopenia (*Lieber, 2010*; *Schiaffino and Reggiani, 2011*; *Buckingham et al., 2003*; *Jungbluth et al., 2005*; *Lawal et al., 2018*; *Nilwik et al., 2013*). Slow twitch muscles (type 1 fibers) are mainly oxidative, contain many mitochondria and are fatigue resistant. Slow twitch muscle, such as soleus, contain muscle fibers expressing the MyHC 1 isoform in addition of muscle fibers expressing MyHC 2 A (*Schiaffino and Reggiani, 2011*). Type 1 fibers are generally less severely affected in patients with neuromuscular disorders such congenital myopathies.

Although such a general classification based on MyHC isoform expression was used for many years by biochemists and physiologists, it has been recently improved thanks to the implementation of 'omic' approaches which have helped refine the phenotypic signature at the single fiber level. A great deal of data has shown that type 2 A fast fibers display a protein profile similar to type I fibers, namely a remarkable level of enzymes involved in oxidative metabolism. Interestingly, type 2 X fibers apparently encode proteins annotated to both oxidative and glycolytic pathways (*Eggers et al., 2021*; *Murgia et al., 2021*).

There are also a number of functionally specialized muscles including extraocular muscles (EOM), jaw muscles and inner ear muscles that have a different embryonic origin and are made up of atypical fiber types (*Schiaffino and Reggiani, 2011*). For example, EOMs are the fastest contracting muscles yet they are fatigue resistant, contain many mitochondria and express most MyHC isoforms including type 1, embryonic and neonatal MyHC as well as EO-MyHC (*Porter et al., 1995*). EOMs are also specifically spared in patients with Duchenne Muscular Dystrophy yet they are affected in patients with some congenital myopathies, including patients with recessive *RYR1* myopathies carrying a hypomorphic or null allele (*Porter et al., 1995*; *Fischer et al., 2002*; *Porter et al., 2003*; *Amburgey et al., 2013*).

Congenital Myopathies (CM) are a genetically heterogeneous group of early onset, non-dystrophic diseases preferentially affecting proximal and axial muscles. More than 20 genes have been implicated in CM, the most commonly affected being those encoding proteins involved in calcium homeostasis and excitation contraction coupling (ECC) and thin-thick filaments (*Jungbluth et al., 2018*). Mutations in *RYR1,* the gene encoding the ryanodine receptor 1 (RyR1) calcium channel of the sarcoplasmic reticulum, are found in approximately 30% of all CM patients, making it the most commonly mutated gene in human CM (*Amburgey et al., 2013*; *Jungbluth et al., 2018*). Within the group of patients carrying *RYR1* mutations, those with the recessive form of the disease are more severely affected, present from birth, have axial and proximal muscle weakness as well as involvement of facial and EOM (*Lawal et al., 2018*; *Amburgey et al., 2013*; *Jungbluth et al., 2018*). A common finding is also the reduced content of RyR1 protein in muscle biopsies (*Zhou et al., 2007*; *Monnier et al., 2008*) which could be one of the causes leading to the weak muscle phenotype. To date, the pathomechanism of disease of recessive *RYR1* mutations is not completely understood and for this reason we created a mouse model knocked in for compound heterozygous mutations identified in a severely affected child with *RYR1*-related congenital myopathy. The double knock in mouse, henceforth referred to as double heterozygous or dHT mouse, carries the RyR1 p.Q1970fsX16 mutation in one allele leading to the absence of a transcript due to nonsense-mediated decay of the allele carrying the frameshift mutation, and the mis-sense RyR1 p.A4329D mutation in the other allele (*Elbaz et al., 2019*). The muscle phenotype of the dHT mouse model closely resembles that of human patients carrying a hypomorphic allele plus a mis-sense *RYR1* mutation, including reduced RyR1 protein content in skeletal muscles, the presence of cores and myofibrillar dis-array, mis-alignment of RyR1 and the dihydropyridine receptor and impaired EOM function (*Elbaz et al., 2019*; *Eckhardt et al., 2020*). Interestingly, beside a reduction in RyR1, the latter muscles also exhibited a significant decrease in mitochondrial number as well

## A. Sample groups

**3 Muscle types:**
- Fast twitch (EDL)
- Slow twitch (soleus)
- Specialized (EOM)

**2 Cohorts:**

**WT** **dHT**

EDL
Soleus
EOM

- Solubilize samples
- LC-MS
- Inject peptides for quantification

## B. Proteomic data analysis

- Quantitative analysis of main proteins involved in ECC and calcium homeostasis.
- Stoichiometry of DHPR to RyR1 and Stim1 to Orai1.

Sarcoplasmic Reticulum

## C. Quantitative analysis

- Statistical analysis on biological replicates.
- Qualitative analysis of up and down-regulated proteins.
- GO pathways analysis.
- Validation based on western blot analysis and published data.

Protein Level

**Figure 1.** Schematic overview of the workflow. (**A**) Skeletal muscles from 12 weeks old WT (5 mice) and dHT littermates (5 mice) were isolated and flash frozen. Three different types of muscles were isolated per mouse, namely EDL, soleus and EOMs. On the day of the experiment, muscles were solubilized and processed for LC-MS. (**B**) For absolute protein quantification, synthetic peptides of RyR1, Cav1.1, Stim1 and Orai1 were used. (**C**) Protein content in different muscle types and in the different mouse genotypes were analyzed and compared.

as changes in the expression and content of other proteins, including the almost complete absence of the EOM-specific MyHC isoform (*Eckhardt et al., 2020*). Such results imply that broad changes in protein expression caused by the mutation and/or reduced content of RyR1 channels, impact other signaling pathways, leading to altered muscle function. A corollary to this is that since not all muscles are equally affected (for example fast twitch muscles and EOMs are more affected than slow twitch muscles) there may be differences in how the *RYR1* mutations affect the different muscle types.

In order to establish how and if *Ryr1* mutations differentially impinge on the expression and function of proteins specific for different muscle types, we performed qualitative and quantitative proteomic analysis of EDL, soleus and EOMs from wild-type and dHT mice.

## Results

*Figure 1* shows a diagram of our experimental workflow: three muscle types were isolated from 12 weeks old wild-type (WT)(n=5) and dHT (n=5) mice, samples were processed for Mass Spectrometry and the results obtained were analyzed against a protein database containing sequences of the predicted SwissProt entries of *Mus musculus* (https://www.ebi.ac.uk/, release date 2019/03/27), *Myh2* and *Myh13* from Trembl, the six calibration mix proteins (*Ahrné et al., 2016*) and commonly observed contaminants (in total 17,414 sequences) using the SpectroMine software. Results obtained from five muscles per group were averaged, filtered so that only changes in protein content ≥0.20 fold and showing a significance of q<0.05 or greater, were considered. In addition, proteins yielding only 1 peptide were not used for analysis and were filtered out.

# Comparison of the proteome of EDL, soleus and EOM muscles from WT mice

In order to perform their specific physiological functions, different muscle types express different protein isoforms or different amounts of specific proteins. For example, slow twitch muscles contain large amounts of the oxygen binding protein myoglobin and of carbonic anhydrase III the enzyme catalyzing the conversion of $CO_2$ to $H_2CO_3$ and $HCO_3^-$ (*Garry et al., 1996*; *Dowling et al., 2021*),

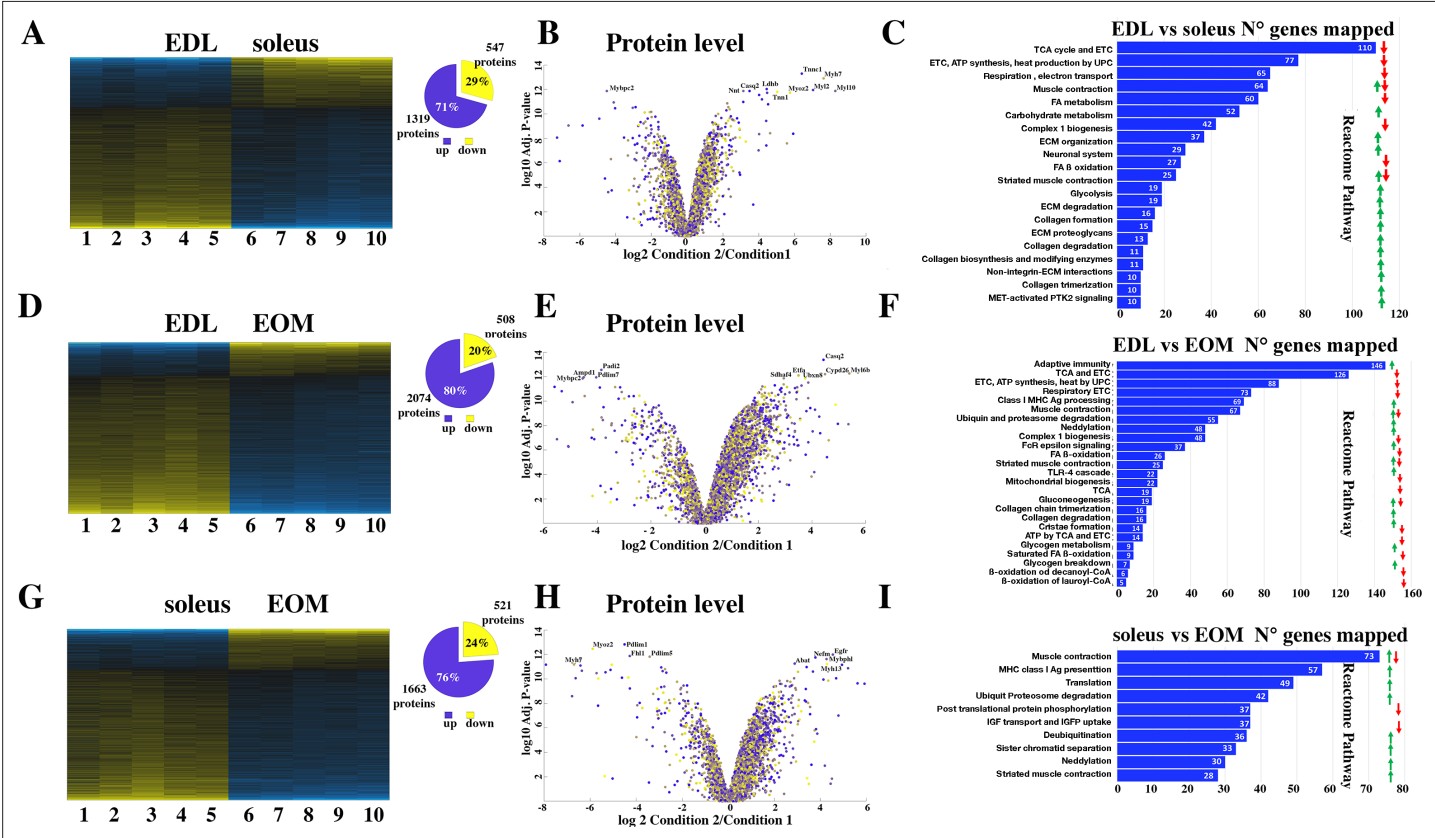

**Figure 2.** Proteomic analysis of EDL, soleus and EOM muscles from WT mice confirms the significant difference in content if proteins involved in the TCA cycle and electron transport chain, fatty acid metabolism and muscle contraction. (**A**) Hierarchically clustered heatmaps of the relative abundance of proteins in EDL (columns 1–5) and soleus muscles (columns 6–10) from five mice. Blue blocks represent proteins which are increased in content, yellow blocks proteins which are decreased in content in EDL versus soleus muscles. Right pie chart shows overall number of increased (blue) and decreased (yellow) proteins. Areas are relative to their numbers. (**B**) Volcano plot of a total of 1866 quantified proteins which showed significant increased (blue) and decreased (yellow) content. The horizontal coordinate is the difference multiple (logarithmic transformation at the base of 2), and the vertical coordinate is the significant difference p value (logarithmic transformation at the base of 10). The proteins showing major change in content are abbreviated. Soleus: condition 2; EDL: condition 1(**C**) Reactome pathway analysis showing major pathways which differ between EDL and soleus muscles. (**D**) Hierarchically clustered heatmaps of the relative abundance of proteins in EDL (columns 1–5) and EOM muscles (columns 6–10) from five mice. Blue blocks represent proteins which are increased in content, yellow blocks proteins which are decreased in content in EDL versus EOM muscles. Right pie chart shows overall number of increased (blue) and decreased (yellow) proteins. Areas are relative to their numbers. (**E**) Volcano plot of a total of 1866 quantified proteins which showed significant increased (blue) and decreased (yellow) content. The horizontal coordinate is the difference multiple (logarithmic transformation at the base of 2), and the vertical coordinate is the significant difference p value (logarithmic transformation at the base of 10). The proteins showing major change in content are abbreviated. EOM: condition 2; EDL: condition 1 (**F**) Reactome pathway analysis showing major pathways which differ between EDL and EOM muscles. (**G**) Hierarchically clustered heatmaps of the relative abundance of proteins in soleus muscles (columns 1–5) and EOM (columns 6–10) from five mice. Blue blocks represent proteins which are increased in content, yellow blocks proteins which are decreased in content in soleus muscles versus EOM. Right pie chart shows overall number of increased (blue) and decreased (yellow) proteins. Areas are relative to their numbers. (**H**) Volcano plot of a total of 1866 quantified proteins which showed significant increased (blue) and decreased (yellow) content. The horizontal coordinate is the difference multiple (logarithmic transformation at the base of 2), and the vertical coordinate is the significant difference p value (logarithmic transformation at the base of 10). The proteins showing major change in content are abbreviated. EOM: condition 2; soleus: condition 1 (**I**) Reactome pathway analysis showing major pathways which differ between soleus and EOM muscles. A q-value of equal or less than 0.05 was used to filter significant changes prior to the pathway analyses. An additional filter was applied to the Heatmaps and Piecharts and only proteins showing a significant change ≥0.2 fold are included.

while fast twitch muscles express large amounts of the calcium buffer protein parvalbumin (*Celio and Heizmann, 1982*) additionally, each muscle type contains specific isoforms of contractile and sarcomeric proteins (*Schiaffino and Reggiani, 2011*). Our first aim was to analyze the proteomes of WT mouse EDL, soleus and EOM muscles to establish their most important qualitative differences (*Figure 2*).

*Figure 2A* shows that the content of more than 1800 proteins are differentially expressed (q<0.05) in soleus compared to EDL muscles from WT mice, of these 547 are present in lower amounts and 1319 are present in higher amounts in soleus compared to EDL muscles; *Figure 2B* shows a volcano plot of the $\log_2$ fold change of proteins in slow (condition 2) versus fast (condition 1) muscles. Reactome pathway analysis (*Figure 2C*) revealed that the pathways showing the greatest number of changes in annotated genes are those encoding proteins associated with mitochondrial function (fatty acid metabolism, TCA cycle, electron transport chain, complex 1 biogenesis, and ß-oxidation) which are significantly reduced in EDL muscles compared to soleus muscles. This is not unexpected considering that slow twitch muscles are made up type I and type IIa/IIx fibers which contain more mitochondria and oxidative enzymes than fast twitch type IIb fibers of fast twitch muscles. On the other hand, EDL muscles are significantly enriched in proteins annotated to muscle contraction, carbohydrate metabolism and glycolysis as well as collagen, integrins and extracellular matrix proteins compared to soleus muscles.

*Figure 2D* shows that the content of more than 2500 proteins are differentially expressed (q<0.05) in EOM compared to EDL from WT mice, of these 508 are present in lower amounts and 2074 are present in higher amounts in EOM compared to EDL muscles. The volcano plot in *Figure 2E* shows the $\log_2$ fold change of proteins in EOM (condition 2) versus fast EDL (condition 1) muscles. Interestingly, Reactome pathway analysis (*Figure 2F*) revealed that EDL muscles contain a larger number of proteins annotated to adaptive immunity and MHC class I antigen presentation compared to EOMs, while the classes of proteins annotated to the citric acid cycle, electron transport chain and fatty acid ß-oxidation are significantly lower in EDL compared to EOMs. This result is in line with the fact that like soleus muscles, or cardiac muscles, EOMs are enriched in mitochondria (*Porter et al., 1995*; *Fischer et al., 2002*) to support continuous movements of the eyes.

*Figure 2H* shows that the content of more than 2000 proteins are differentially expressed (q<0.05) in EOM compared to soleus from WT mice, of these 521 are present in lower amounts and 1663 are present in higher amounts in EOM compared to soleus muscles. The volcano plot in *Figure 2H* shows the $\log_2$ fold change of proteins in EOM (condition 2) versus slow soleus (condition 1) muscles. Reactome pathway analysis (*Figure 2I*) revealed that the most affected category is that containing genes annotated to muscle contraction (that were both up- and downregulated) followed by genes involved in MHC class I antigen presentation, translation and ubiquitin/proteasome degradation that are upregulated in soleus muscles compared to EOM muscles. Reactome pathway analysis as well as Genome Ontology pathway analysis are not sufficiently informative and probably miss important groups of proteins specific to skeletal muscle function; this observation prompted us to select specific proteins whose expression level is known to be different between fast, slow and EOM muscles. Focusing on the relative change in protein content between EDL and soleus muscles of contractile and sarcomeric proteins, our results confirm that the slow muscle Troponin I and C1 isoforms as well as the slow-MyHC 1 (encoded by *Myh7*) are enriched between 32 and 197-fold in soleus muscles, whereas α-actinin 3 and 4 and myomesin 1 are more abundant in EDL muscles and desmin is enriched in soleus muscles (*Supplementary file 1a*). Analysis of sarcoplasmic reticulum proteins involved in ECC show that the content of calsequestrin 2 and SERCA2 is 11- and 22-fold higher in soleus muscles, whereas the relative content of proteins of the junctional face membrane of the sarcoplasmic reticulum involved in ECC (*Treves et al., 2009*) including RyR1, the dihydropyridine (DHPR) complex (including the α1, β1, and α2δ subunits), Stac3, junctophilin-1 and triadin is more than 50% higher in EDL muscles compared to soleus, as is FKBP12 which binds to and stabilizes the RyR1 complex (*Brillantes et al., 1994*). Fast twitch muscles are also enriched in SERCA1, calsequestrin 1 and junctophilin 2. The abundance of protein annotated to calcium signaling and sarcoplasmic reticulum in EDL is consistent with the larger membrane volume of sarcotubular membrane in fast-twitch muscles compared to slow twitch muscles (*Luff and Atwood, 1971*).

A similar approach was used to compare the relative content of specific proteins changing between EDL and EOMs and soleus and EOMs. Importantly, the results of the mass spectroscopy approach

reported here validate a great deal of experimental observations including the fact that EOMs express high levels of *Myhc13*, a specific extra-ocular muscle MyHC isoform (MyHC-EO), as well as more cardiac muscle specific protein isoforms. For example, within the contractile and sarcomeric protein category, compared to EDL muscles, EOMs are particularly enriched in MyHC-slow (24-fold), MyHC-EO (29-fold) and Troponin C1 (slow and cardiac muscle isoform, 31-fold), whereas they contain very low amounts of α-actinin 3 (0.02-fold), MyHC 2b (0.07-fold) and MyHC 2 X (0.61-fold). Within the ECC coupling category, EOMs are enriched in calsequestrin 2 (21-fold), SERCA2 (3.6-fold) and junctin/junctate/aspartyl-ß-hydroxylase (3.5-fold) whereas their content of RyR1, the α-1 subunit of the dihydropyridine receptor (DHPRα1s), calsequestrin 1, Stac3, junctophilin-1 and triadin is significantly reduced by more than 50% compared to EDL muscles (*Supplementary file 1b*). Similarly, soleus muscles and EOMs vary in their content of a large number of proteins. Within the contractile and sarcomeric protein category, EOMs are enriched in embryonic MyHC (*Myhc3*, 49-fold), MyHC-EO (20-fold) and cardiac troponin T (3.10-fold), whereas compared to soleus muscles they contain very low amounts of slow- MyHC (0.0096-fold), myosin light chain 2 (0.01-fold), myozenin-2 (0.017-fold) and α-actinin 2 (0.017-fold). In the ECC category, EOMs are enriched in a number of proteins including SERCA1 (eight-fold) and SERCA3 (sevenfold), Stim1 (fourfold), Junctin/junctate/Aspartyl-ß-hydroxylase (threefold), DHPRα1s (1.4-fold) and junctophilin-1 (1.4-fold), whereas they contain very low amounts of SERCA2 and >50% lower amounts of Mitsugumin 53 (*Supplementary file 1c*). Interestingly compared to EDL and soleus muscles, EOMs are enriched more than twofold in Stim1, junctin/junctate/aspartylß-hydroxlase. Furthermore, compared to soleus muscles and EOMs, EDLs are enriched in parvalbumin and in proteins annotated to calcium-dependent signaling' via the calcium /calmodulin dependent protein kinase IIα and IIγ, whereas soleus and EOM muscles are enriched in S100A1.

Altogether, the results of the mass spectrometry analysis not only confirm known differences between muscle types (*Schiaffino and Reggiani, 2011*; *Porter et al., 1995*; *Fischer et al., 2002*; *Celio and Heizmann, 1982*; *Luff and Atwood, 1971*) but also reveal new molecular signatures of EDL, soleus and EOMs. In this context, it is worth mentioning that more than 10 heat shock proteins are more abundant in soleus muscles and EOMs compared to EDL muscles, including *Hspb6* (16-fold higher in soleus compared to EDL) and *Hspa12a* (7-fold higher in EOM vs soleus). *Hspb6* has been implicated in protection against atrophy, ischemia, hypertensive stress, and metabolic dysfunction (*Dreiza et al., 2010*). Importantly, a great deal of data has shown that muscles from patients with several neuromuscular disorders including those caused by *RYR1* mutations show fiber type 1 predominance (*Jungbluth et al., 2005*; *Lawal et al., 2018*) and heat shock proteins have been suggested to have a protective effect against muscle damage caused by calcium dysregulation and uncoupling of mitochondrial respiratory chain (*Maglara et al., 2003*) as well as protective effects against ischemic injury in cardiomyocytes (*Martin et al., 1997*). Interestingly, the content of Mitsugumin 53 (encoded by *Trim72*), a protein involved in muscle membrane repair (*Cai et al., 2009*) is 2.8-fold higher in slow twitch muscles compared to fast twitch muscles. Thus, on the basis these observations we cannot exclude the possibility that increased expression of Mitsugumin 53, along with a set of heat shock proteins (*Dreiza et al., 2010*; *Maglara et al., 2003*; *Martin et al., 1997*; *Larkins et al., 2012*), might be relevant in preventing muscle fiber type 1 damage associated with the presence of recessive *RYR1* mutations or with other type of stressing events. To directly verify this hypothesis, we examined the proteome of fast and slow twitch muscles in a mouse model (RyR1 dHT) of congenital muscle disorders carrying the p.Q1970fsX16 mutation in one allele and the mis-sense p.A4329D mutation in the other allele (*Elbaz et al., 2019*).

## Comparison of muscles isolated from WT and RyR1 dHT mice

In the next experiments, the proteome of three different muscles from dHT mice vs those of WT mice were compared. *Figure 3A and B* shows that in EDL muscles a total of 848 proteins are significantly (q<0.05) mis-regulated in dHT mice; in particular, 529 and 319 proteins are up- or downregulated only in the EDLs of dHT mice compared to WT mice, respectively. Reactome pathway (*Figure 3—figure supplement 1A*) analysis revealed that proteins involved in homeostasis of the extracellular matrix, including collagen assembly and chain formation, collagen degradation, ECM organization and integrin interaction, are up-regulated in EDLs from WT compared to dHT mice. We also compared the proteome of soleus muscles from WT and dHT mice. *Figure 3C and D* show that the overall number of proteins showing significant changes in their relative content between dHT and WT mice, is smaller

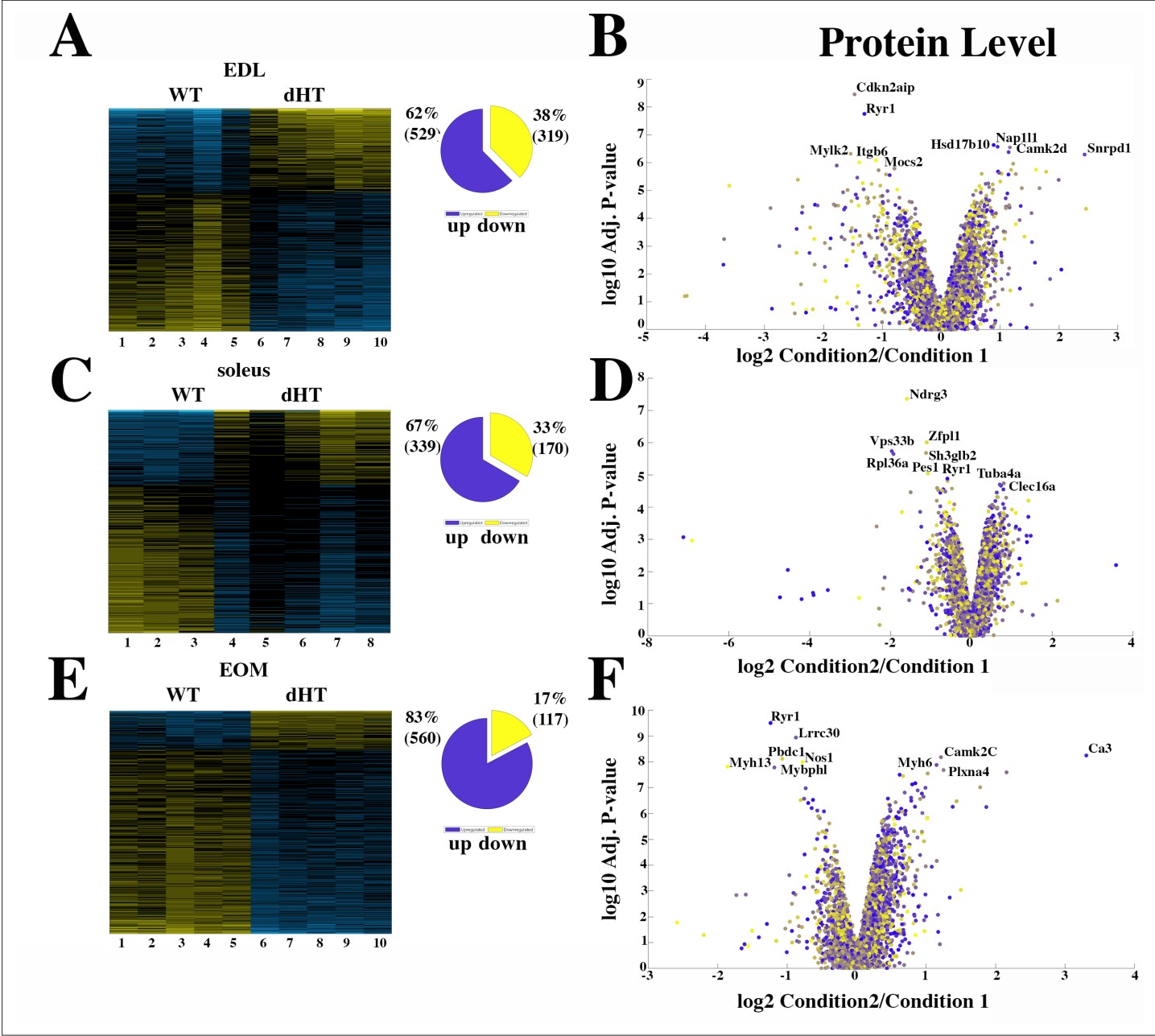

**Figure 3.** Proteomic analysis comparison of muscles from dHT and WT mice. (**A, C and E**) Hierarchically clustered heatmaps of the relative abundance of proteins in EDL (**A**), soleus muscles (**C**) and EOMs (**E**) from three to five mice. Blue blocks represent proteins which are increased in content, yellow blocks proteins which are decreased in content in WT (columns 1–5 in A and E; 1–3 in C) versus dHT (5–10 in A and E; 4–8 in C). Right pie chart shows overall number of increased (purple) and decreased (yellow) proteins. Areas are relative to their numbers. (**B, D and F**) Volcano plots of total quantified proteins showing significant increased (blue) and decreased (yellow) content in dHT (condition 2) versus WT (condition 1) EDL (**B**), soleus (**D**) and EOMs (**F**). The horizontal coordinate is the difference multiple (logarithmic transformation at the base of 2), and the vertical coordinate is the significant difference p value (logarithmic transformation at the base of 10). The proteins showing major change in content are abbreviated. A q-value of equal or less than 0.05 was used to filter significant changes prior to the pathway analyses. An additional filter was applied to the Heatmaps and Piecharts and only proteins showing a significant change ≥0.2-fold are included.

The online version of this article includes the following figure supplement(s) for figure 3:

**Figure supplement 1.** Reactome pathway analysis showing major pathways which differ between EDL muscles (**A**) and EOM muscles (**B**) in dHT versus WT mice.

than that observed in EDL muscles. In particular, we found that 339 and 170 proteins are up- or downregulated only in the soleus muscles of dHT mice compared to those from WT mice, respectively. Contrary to EDL muscles, Reactome pathway analysis failed to identify a preferentially affected cellular pathway.

Since ophthalmoplegia is a common clinical sign observed in patients affect by congenital myopathies linked to recessive *RYR1* mutations (*Lawal et al., 2018*; *Amburgey et al., 2013*; *Jungbluth et al., 2018*), we also investigated the proteome of EOMs from dHT and WT mice. *Figure 3E and F* shows that 560 and 117 proteins are up- or downregulated only in the EOM of dHT mice compared to WT mice, respectively. Interestingly, Reactome pathway analysis indicated that genes encoding proteins involved in the citric acid cycle and electron transport chain, ATP synthesis and uncoupling protein complexes linked to heat formation are upregulated in dHT vs WT EOMs (*Figure 3—figure supplement 1B*).

The Venn diagram (*Figure 4*) shows that the three muscle types from the dHT mice share a number of proteins whose content increases or decreases. It also shows that there are a number of proteins whose content increases or decreases in a specific muscle type only, namely 848 proteins in EDL, 677 proteins in soleus and 509 proteins in EOMs. We analyzed these proteins to verify whether they were annotated to specific cellular pathways but the results were not sufficiently informative as far as skeletal muscle function, ECC and calcium homeostasis are concerned. In fact, GO analysis showed that the genes encoding the proteins that were downregulated or upregulated specifically in dHT EDL, soleus and EOM were annotated to the Biological processes category, specifically: biological cellular processes, response to stimulus and multicellular organismal process (downregulated) and metabolic process, response to stimulus and positive regulation of biological process (upregulated) in EDL muscles from dHT mice (*Figure 4—figure supplement 1A and D*), cellular process, biological regulation and metabolic process (downregulated) and cellular process, metabolic process and organic substance metabolic process (upregulated) in soleus muscles from dHT mice (*Figure 4—figure supplement 1B and E*) and cellular process, cellular metabolic process and oxidation reaction process (downregulated) and cellular process, primary metabolic process and regulation of biological quality (upregulated) in EOM muscles from dHT mice (*Figure 4—figure supplement 1C and F*).

Thus, we selected and analyzed protein families playing a role in skeletal muscle ECC, muscle contraction, collagen and ECM, heat shock response/chaperones, protein synthesis and calcium-dependent regulatory functions that exhibit a significantly different ($q < 0.05$) content between the two mouse genotypes. *Table 1* shows that several proteins involved in skeletal muscle ECC are downregulated in EDLs from dHT mice, including the RyR1 (60% decrease) as well as its stabilizing binding protein FKBP12, DHPRα1 and junctophillin 1 whose relative content decreases by more than 30%, 23%, and 40%, respectively. The content of Asph which encodes different proteins including junctin, junctate, humbug and aspartyl-ß-hydroxylase (*Treves et al., 2000*) increases almost twofold, whereas SRP-35 (*Dhrs7c*) increases 1.34-fold in EDLs from dHT mice. Additionally, the expression of type 2 fibers is impacted since MyHC 2 X and 2B as well asα-actinin 3 (which is preferentially expressed in type 2 fibers) (*Schiaffino and Reggiani, 2011*) are decreased in the EDLs of dHT mice. The decrease of the fast isoforms of MyHC in dHT EDL muscles is accompanied by a decrease of many collagen isoforms. On the other hand, the content of several heat shock proteins as well as the content of 60 S and 40 S ribosomal proteins is increased in fast twitch fibers from the dHT. In addition, we found that the calcium/CaM-dependent protein kinases 1, 2α 2β and 2δ are increased in EDL from dHT mice.

We applied a similar approach as described above (i.e. proteins showing significantly different ($q < 0.05$)≥0.2-fold change in content between the two mouse genotypes) to identify important components differing between dHT and WT soleus muscles (*Table 2*). In the ECC protein category, RyR1, DHPRα1s, and Junctophillin 1 are significantly decreased, as is triadin, whereas junctin/junctate/ß-hydroxylase and SERCA2 are increased in muscles from dHT mice. In the contractile protein group, significant changes are only observed for Troponin 3 whose content decreases by about 30%. Similar to what was observed in EDL muscles, we found that the content of calcium/calmodulin-dependent protein kinases IIδ and γ is increased. In addition, S100A1, a calcium binding protein which binds to and regulates RyR1 activity (*Treves et al., 1997*; *Prosser et al., 2011*), is significantly increased in soleus muscles from dHT mice. Finally, proteins constituting the 60 S and 40 S ribosomal subunits are increased in soleus muscles from dHT mice compared to WT.

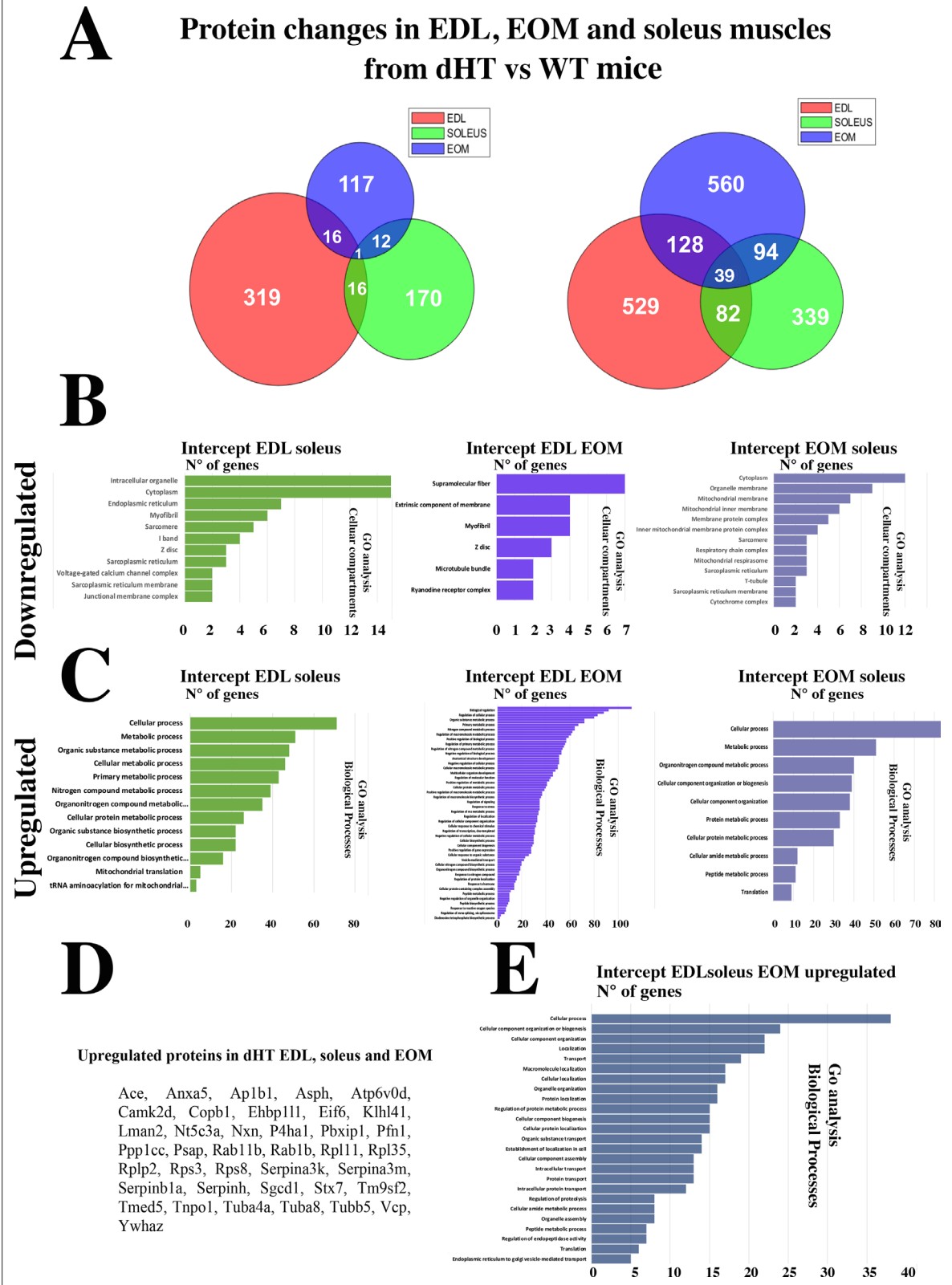

**Figure 4.** Changes in protein content in EDL, soleus and EOM between dHT vs WT mice. (**A**) Venn diagram showing significantly decreased proteins (left) and increased proteins (right) in the three muscle types. (**B**) GO biological process analysis of common proteins that are downregulated and (**C**) upregulated in muscle from dHT mice. Left panels, common proteins showing significant changes in content in both EDL and soleus muscles. Central panels, common proteins showing significant changes in content in EDL and EOMs; right panels, common proteins showing significant changes

*Figure 4 continued*

in content in EOM and soleus muscles. (**D**) List of the 39 proteins whose content is increased in EDL, soleus and EOMs in dHT mice. (**E**) GO analysis annotated to Biological processes of the 39 proteins that are increased in muscles from dHT mice.

The online version of this article includes the following figure supplement(s) for figure 4:

**Figure supplement 1.** Gene Ontology annotated to Biological process genes showing significant differences in content between muscles from dHT and WT mice.

In EOMs, the proteins showing the greatest fold change (aside those involved in ECC and muscle contraction), are heat shock proteins, ribosomal proteins and proteins of the ECM, a variety of heat shock proteins and calcium/calmodulin-dependent protein kinases IIβ and IIδ and S100 family proteins (*Table 3*).

Of note, the overall changes caused by *Ryr1* mutations on the protein composition of muscles is more prominent in fast twitch muscles such as EOMs and EDLs compared to the slow twitch soleus muscle and the proteomic approach revealed that the content of many proteins differs between dHT and WT EDL, soleus and EOM muscles. We next refined our analysis and searched for protein whose content variation is most strongly associated with the dHT genotype. In particular, we searched for proteins which show significant changes in content in all three muscle types, namely EDL, Sol, and EOM. The Venn diagram (*Figure 4*) shows that the three muscle types from the dHT mice share a number of proteins whose content increases or decreases. The downregulation of RyR1 appears to be a unique a signature of the dHT phenotype, since its decrease is the only change shared between all muscle types investigated (*Figure 4A*). Biological process GO analysis revealed that the content of genes annotated to intracellular organelles, cytoplasm and ER (*Figure 4B* left panel, downregulated) and cellular process, metabolic process and organic substance metabolic process (*Figure 4C* left panel, upregulated) amongst others is changed in both EDL and soleus muscles from dHT mice. On the other hand, the content of genes annotated to supramolecular fiber, extrinsic component of membrane and myofibril (*Figure 4B* middle panel, downregulated), biological regulation, regulation of cellular process and organic substance metabolic process (*Figure 4C* middle panel, upregulated) amongst others is changed in both EDL and EOMs from dHT mice. While the content of genes annotated to cytoplasm, organelle membrane and mitochondrial membrane (*Figure 4B* right panel, downregulated) and cellular process, metabolic process and organonitrogen compound metabolic process (*Figure 4C* right panel, upregulated) amongst others is changed in both soleus and EOMs from dHT mice. Two heat shock proteins Hsp70 (BiP) and Hsp family B small member 6 (*Hspb6*) are increased only in EDL and EOMs. Interestingly, the content of 39 proteins including Kelch-like protein 41, two annotated to calcium signaling such as calmodulin kinase 2δ and aspartyl-ß -hydroxylase are increased in all three muscle types from dHT mice (*Figure 4A and D*), as are several proteins associated with the 40 S and 60 S ribosomal subunits. GO analysis of biological process revealed that the content of genes annotated to cellular process, cellular component organization or biogenesis and cellular component organization amongst others are upregulated in the three muscle types in dHT mice (*Figure 4E*).

## Quantification and stoichiometry of ECC proteins in WT and dHT muscles

Skeletal muscle ECC relies on the highly ordered architecture of two intracellular membrane compartments, namely the transverse tubules which are invaginations of the plasma membrane containing the DHPR macromolecular complex and the sarcoplasmic reticulum containing the RyR1 macromolecular complex, as well as other proteins involved in calcium homeostasis and accessory structural proteins (*Treves et al., 2009*; *Franzini-Armstrong and Jorgensen, 1994*). The relative content of many of these proteins has been determined, nevertheless few studies have established their stoichiometry in relation to particular muscle types (*Franzini-Armstrong and Jorgensen, 1994*; *Smith et al., 2013*; *Leberer and Pette, 1986*). Within a total muscle homogenate, sarcoplasmic reticulum membrane proteins are of low abundance, thus, to quantify these proteins we performed high-resolution TMT mass spectrometry by using spiked-in labeled peptides from major protein involved in key steps of ECC calcium signaling to build a standard calibration curve. In particular, we used peptides from RyR1 and DHPRα1s, and Stim1 and Orai1, proteins which are involved in calcium release from the SR and in calcium entry across sarcolemma, respectively. The obtained protein concentrations

**Table 1.** Relative change in the content of selected proteins in EDL muscles isolated from WT (baseline) and dHT mice.

| | Gene name | Protein* | Relative content | q value |
|---|---|---|---|---|
| | *Ryr1* | Ryanodine receptor 1 (RyR1) | 0.40 | $3.97 \times 10^{-5}$ |
| | *Jph1* | Junctophillin-1 | 0.64 | 0.025 |
| | *Cacna1s* | Voltage dependent L type calcium channel subunit a1s (DHPR α1s) | 0.73 | 0.018 |
| | *Dhrs7c* | Dehydrogenase/reductase SDR family member 7 C (SRP-35) | 1.34 | 0.0045 |
| ECC | *Asph* | Aspartyl/asparaginyl ß-hydroxylase (junctin/junctate/asp-ß-hydroxylase) | 1.84 | 0.00095 |
| | *Myh13* | MyHC-EO | 0.35 | 0.0063 |
| | *Myh1* | Myosin-1 (MyHC-2x) | 0.61 | 0.043 |
| | *Myh4* | Myosin-4 (MyHC 2b) | 0.71 | 0.018 |
| Contractile proteins | *Actn3* | α-actinin 3 | 0.74 | 0.012 |
| | *Col2a1* | Collagen (II)α–1 chain | 0.18 | 0.0043 |
| | *Col1a2* | Collagen (I) α –2 chain | 0.25 | 0.027 |
| | *Col11a1* | Collagen (XI) α –1 chain | 0.35 | 0.0047 |
| | *Col5a2* | Collagen (V)α –2 chain | 0.37 | 0.00059 |
| | *Col5a1* | Collagen (V) aα –1 chain | 0.50 | 0.00156 |
| | *Col16a1* | Collagen (XVI) α –1 chain | 0.53 | 0.00154 |
| | *Col4a2* | Collagen (IV) α –2 chain | 0.7 | 0.040 |
| | *Itgav* | Integrin α -V | 0.77 | 0.044 |
| Collagen and ECM proteins | *Itgb1bp2* | Integrin ß–1- binding protein 2 | 1.3 | 0.045 |
| Heat shock proteins | *Hspb3* | Hsp ß–3 | 0.73 | 0.00376 |
| | *Hspb8* | Hsp ß–8 (a-crystallin C chain) | 0.75 | 0.0160 |
| | *Hspa2* | Heat shock related 70 kDa protein (Hsp70-2) | 0.77 | 0.026 |
| | *Hspd1* | 60 kDa Hsp, mitochondrial (Chaperonin 60) | 1.30 | 0.011 |
| | *Hspa5* | Heat Shock Protein Family A (Hsp70) Member 5 (BiP) | 1.41 | 0.00928 |
| | *Hsph1* | Hsp 105 kDa (Hsp105, Hsp110) | 1.47 | 0.0155 |
| | *Hspb6* | Hsp ß–6 (HspB6) | 1.5 | 0.0259 |
| | *Hspbp1* | Hsp 70-binding protein | 1.8 | 0.022 |

*Table 1 continued on next page*

*Table 1 continued*

|  | Gene name | Protein* | Relative content | q value |
|---|---|---|---|---|
|  | *Rpl23* | 60 S Ribosomal protein L23 | 0.433 | 0.023 |
|  | *Mrpl1* | 39 S ribosomal protein L1, mitochondrial | 0.526 | 0.004 |
|  | *Mrpl46* | 39 S ribosomal protein L46, mitochondrial | 0.592 | 0.011 |
|  | *Rpl34* | 60 S ribosomal protein L34 | 0.659 | 0.0042 |
|  | *Rps15a* | 40 S ribosomal protein S15a | 0.659 | 0.0056 |
|  | *Mrpl43* | 39 S ribosomal protein L43, mitochondrial | 0.684 | 0.029 |
|  | *Mrps5* | 28 S ribosomal protein S5, mitochondrial | 0.74 | 0.0021 |
|  | *Rpl11* | 60 S ribosomal protein L11 | 1.265 | 0.013 |
|  | *Rpl6* | 60 S ribosomal protein L6 | 1.273 | 0.012 |
|  | *Rpl35* | 60 S ribosomal protein L35 | 1.290 | 0.034 |
|  | *Mrpl19* | 39 S ribosomal protein L19, mitochondrial | 1.346 | 0.028 |
|  | *Rps25* | 40 S ribosomal protein S25 | 1.35 | 0.0076 |
|  | *Rpl27a* | 60 S ribosomal protein L27a | 1.365 | 0.02 |
|  | *Rpl27* | 60 S ribosomal protein L27 | 1.374 | 0.016 |
|  | *Rpl9* | 60 S ribosomal protein L9 | 1.374 | 0.015 |
|  | *Rps2* | 40 S ribosomal protein S2 | 1.39 | 0.0065 |
|  | *Rps8* | 40 S ribosomal protein S8 | 1.39 | 0.013 |
|  | *Rplp2* | 60 S acidic ribosomal protein P2 | 1.403 | 0.0087 |
|  | *Rps10* | 40 S ribosomal protein S10 | 1.41 | 0.033 |
|  | *Rpl38* | 60 S ribosomal protein L38 | 1.431 | 0.025 |
|  | *Rpl23a* | 60 S ribosomal protein L23a | 1.459 | 0.005 |
|  | *Rps12* | 40 S ribosomal protein S12 | 1.473 | 0.0128 |
|  | *Rps9* | 40 S ribosomal protein S9 | 1.50 | 0.0278 |
|  | *Rpl18* | 60 S ribosomal protein L18 | 1.491 | 0.017 |
|  | *Mrps7* | 28 S ribosomal protein S7, mitochondrial | 1.567 | 0.010 |
|  | *Rpl10a* | 60 S ribosomal protein L10a | 1.591 | 0.017 |
|  | *Rpl22* | 60 S ribosomal protein L22 | 1.651 | 0.017 |
|  | *Rps17* | 40 S ribosomal protein S17 | 1.661 | 0.0016 |
| Ribosomal proteins | *Rps16* | 40 S ribosomal protein S16 | 1.82 | 0.0020 |
|  | *Fkbp1a* | Peptidyl-prolyl cis-trans isomerase FKBP1A (FKBP12; calstabin-1) | 0.64 | 0.0025 |
|  | *Fkbp8* | Peptidyl-prolyl cis-trans isomerase FKBP8 (38 kDa FKBP) | 1.30 | 0.024 |
| FK506 binding proteins | *Fkbp9* | Peptidyl-prolyl cis-trans isomerase FKBP9 (63 kDa FK506-binding protein) | 1.60 | 0.0057 |
| Calcium-dependent protein kinases | *Camk1* | Calcium/Calmodulin-dependent protein kinase type 1 (CaM kinase I) | 1.32 | 0.022 |
|  | *Camk2a* | Calcium/calmodulin-dependent protein kinase type II subunit α | 1.40 | 0.0189 |
|  | *Camk2b* | Calcium/calmodulin-dependent protein kinase type II subunit ß | 1.46 | 0.010 |
|  | *Camk2d* | Calcium/calmodulin-dependent protein kinase type II subunit δ | 2.24 | 0.00025 |

*Table 1 continued*

| | Gene name | Protein* | Relative content | q value |
|---|---|---|---|---|
| | *Psmd7* | 26 S proteasome non-ATPase regulatory subunit *7* | 0.58 | 0.0016 |
| | *Psmg2* | Proteasome assembly chaperone 2 | 1.66 | 0.038 |
| Varia | *Fth1* | Ferritin | 1.69 | 0.0033 |

*The nomenclature of Proteins is based on that of the UniProtKB database.

showed a high correlation (R2=0.96; *Figure 5*) with the MS abundance estimates determined from the global proteomics analysis. Therefore, we used this curve to extrapolate the absolute amounts and stoichiometry of proteins whose values fall within the linear domain of the curve, namely, JP-45, triadin, junctophilin 1, Stac3 in addition to RyR1, DHPRα1s, Stim1 and Orai1. The content of the RyR1 protomer in WT fast twitch EDL muscles is 1.29±0.07 µmol/kg wet weight, whereby the calculated RyR1 tetrameric complex is 0.32 µmol/kg wet weight (*Table 4*) a value which is threefold lower compared to that determined in total muscle homogenates by[³H]-ryanodine equilibrium binding by Bers et al. (*Bers and Stiffel, 1993*). On the other hand, our RyR1 quantification results in mouse total muscle homogenates obtained by TMT mass spectrometry using labeled peptides is approximately fivefold higher compared to those obtained in rabbit and frog whole skeletal muscle homogenate preparations by *Anderson et al., 1994* and by *Margreth et al., 1993*. Our results also show that the RyR1 concentration (in µmol/Kg) in soleus and EOM muscles from WT mice is approximately 38% and 46% of that found in EDL muscles of WT mice, respectively (*Table 4*). We found that the content of DHPRα1s in EDL muscles is 0.56±0.03 mol/kg wet weight, a value approx. 2.5-fold higher compared to that of soleus muscles (0.18±0.01 µmol/kg wet weight) and of EOMs (0.21±0.01 mol/kg wet weight). Thus, the calculated RyR1 tetramer to DHPRα1s ratio in EDL muscles from WT and dHT mice is 0.571 and 0.429, respectively (*Table 5*). Such a value appears to be slightly higher both in soleus and EOM muscles (0.667 and 0.625 in WT and dHT soleus muscles and 0.714 and 0.474 in WT and dHT EOMs, respectively, *Table 5*). The Stac3 content correlates with that of DHPRα1s, namely EDL is the muscle which is most enriched in Stac3 (0.62±0.07 µmol/kg wet weight); soleus and EOMs contain approximately one-third of the Stac 3 present in the EDL, namely 0.22±0.02 mol/kg wet weight and 0.17±0.01 mol/kg wet weight in soleus and EOMs, respectively. Stac3 content in muscles from dHT is similar to that of WT littermates. Interestingly, the content of Stim1 depends on the muscles type. Mass spectrometry quantification revealed that EOMs contain the highest amounts of Stim1 (1.35±0.03 µmol/kg wet weight) compared to soleus (0.55±0.03 µmol/kg wet weight) and EDL (0.46±0.02 µmol/kg wet weight). Western blot analysis of total muscle homogenates from WT mice confirmed that EOMs contain four times more Stim1 than EDL muscles *Figure 6* and *Supplementary file 1b* and that equal proportions of Stim1 and Stim1L are present in the three muscle groups, with no preferential expression of the long isoform in any of the muscles investigated. As to WT EDL and soleus, we found no major differences in Stim1 expression, confirming previous data by *Cully et al., 2016*. The expression of Stim1 is accompanied by the expression of Orai1 in EDL and EOMs but not in soleus muscles. Indeed, mouse EOMs contain the highest amount of Orai1 monomer (0.16±0.03 µmol/kg wet weight) and EDLs contained approximately 68% of that (0.11±0.01 µmol/kg wet weight). To our surprise in soleus muscles, the content of Orai1 is below the detection level of mass spectrometry measurement, indicating that slow twitch (soleus) muscles express very little, if any, Orai1 compared to fast twitch EDL and EOMs.

## Discussion

To understand in greater detail the changes in skeletal muscle function in congenital myopathies caused by recessive *RYR1* mutations, we performed an in-depth qualitative and quantitative analysis of protein content and abundance in EDL, soleus and EOMs from WT and dHT mice. The results of the proteomic analysis reveal that, asides the drastic reduction in RyR1 content, profound changes occur in the content of many proteins particularly in fast-, slow-twitch and EOM muscles. Namely, we found that recessive *Ryr1* mutations lead to an increase content aspartyl-ß-hydroxylase (Asph), some ribosomal proteins and calmodulin kinase 2 delta. EDL and EOMs that are more severely affected, also

**Table 2.** Relative change in the content of selected proteins in soleus muscles isolated from WT (baseline) and dHT mice.

| | Gene name | Protein* | Relative content | q value |
|---|---|---|---|---|
| ECC | Ryr1 | Ryanodine receptor 1 (RyR1) | 0.66 | 0.0080 |
| | Jph1 | Junctophillin 1 | 0.73 | 0.026 |
| | Cacna1s | Voltage-dependent L type calcium channel subunit α1s (DHPR α1s) | 0.67 | 0.017 |
| | Trdn | Triadin | 0.69 | 0.0352 |
| | Asph | Aspartyl/asparaginyl ß-hydroxylase (junctin/junctate/aspß-hydroxylase) | 1.31 | 0.045 |
| | ATP2a2 | Sarcoplasmic/endoplasmic reticulum calcium ATPase 2 (SERCA2) | 1.65 | 0.0256 |
| Contractile proteins | Tnnt3 | Troponin 3 (fast skeletal muscle type) | 0.69 | 0.017 |
| Calcium binding proteins | S100a1 | Protein S100 A1 | 1.69 | 0.033 |
| | Camk2d | Calcium/calmodulin-dependent protein kinase type II subunit δ | 1.23 | 0.033 |
| | Camk2g | Calcium/calmodulin-dependent protein kinase type II subunitγ | 1.43 | 0.039 |
| Ion Pumps | Atp1b1 | Na+/K+ATPase ß1 | 0.77 | 0.047 |
| | Atp1a1 | Na+/K+ATPase α 1 | 1.41 | 0.017 |
| Collagen and ECM proteins | Col11a1 | Collagen (XI) α −1 chain | 1.63 | 0.031 |
| | Itgb5 | Integrin a V/ß−5 | 11.95 | 0.044 |
| Ribosomal proteins | Rpl36a | 60 S ribosomal protein L36a | 0.263 | 0.0021 |
| | Mrpl10 | 39 S ribosomal protein L10, mitochondrial | 0.577 | 0.040 |
| | Rpl8 | 60 S ribosomal protein L8 | 0.661 | 0.022 |
| | Rpl26 | 60 S ribosomal protein L26 | 0.695 | 0.022 |
| | Mrpl42 | 39 S ribosomal protein L42, mitochondrial | 0.788 | 0.026 |
| | Rpl30 | 60 S ribosomal protein L30 | 1.259 | 0.020 |
| | Rpl19 | 60 S ribosomal protein L19 | 1.260 | 0.050 |
| | Mrpl41 | 39 S ribosomal protein L41, mitochondrial | 1.289 | 0.033 |
| | Rpl11 | 60 S ribosomal protein L11 | 1.325 | 0.026 |
| | Rpl10 | 60 S ribosomal protein L10 | 1.432 | 0.013 |
| | Rpl22 | 60 S ribosomal protein L22 | 1.436 | 0.017 |
| | Rplp2 | 60 S acidic ribosomal protein P2 | 1.473 | 0.022 |
| | Rpl35 | 60 S ribosomal protein L35 | 1.533 | 0.040 |
| | Rpl23a | 60 S ribosomal protein L23a | 1.612 | 0.014 |
| | Rpl23 | 60 S ribosomal protein L23 | 1.638 | 0.022 |

*Table 2 continued on next page*

*Table 2 continued*

|  | Gene name | Protein* | Relative content | q value |
|---|---|---|---|---|
| Varia | *Psmg1* | Proteasome Assembly Chaperone 1 | 0.488 | 0.024 |
|  | *Dnajb6* | DnaJ homolog subfamily B member 6 (Hsp J-2) | 1.45 | 0.033 |
|  | *Psma2* | Proteasome 20 S Subunit α 2 | 1.51 | 0.011 |

*The nomenclature of Proteins is based on that of the UniProtKB database.

share changes in the content of other proteins, including collagens, heat shock proteins, and CamK2b as well as additional ribosomal proteins. We believe that the reduced RyR1 calcium channel content has a domino effect leading to changes in content of many proteins, particularly in EDL and EOMs.

## Stoichiometry of ECC molecular complex in health and diseased muscles

In this study we used spiked-in labelled peptides for isobaric TMT mass spectrometry measurements to quantify the major protein components of the ECC molecular complex in EDL, soleus and EOM from WT and dHT mice and established the absolute content of low abundant ECC-molecular complex proteins, including RyR1, DHPRα1s, Stim1, Orai1. The calculated values for RyR1 and DHPRα1s that we obtained are of the same order of magnitude as those previously determined by equilibrium ligand binding (*Anderson et al., 1994*; *Margreth et al., 1993*; *Cully et al., 2016*), confirming the reliability of this approach. In addition, we also provide for the first time the absolute quantification of Stim1 and Ora1, two crucial proteins involved in Store Operated Calcium Entry (SOCE). Our results are interesting because of the widespread attention gained by SOCE in skeletal muscle, not only because mutations in *STIM1* and *ORAI1* are the underlying feature of several genetic diseases associated with muscle weakness (*Böhm et al., 2013*; *Lacruz and Feske, 2015*), but also because experimental evidence has shown that Stim1 and Orai1 play an important role in refilling intracellular calcium stores in fast and slow twitch muscles (*Cully et al., 2016*; *Wei-Lapierre et al., 2013*; *Carrell et al., 2016*). Quantitative isobaric TMT mass spectrometry revealed important differences in the content of Stim1, Stim2 and Orai1 among different muscle types. We are confident of our results because the data relative to Stim1 content were validated by staining western blots of total muscle homogenates with Stim1-specific antibodies. Our data show that EOMs contain the highest levels of Stim1, and, in agreement with previous data by *Cully et al., 2016*, we found no major differences in Stim1 content between fast and slow twitch muscles. The Stim1 to Orai1 ratio in EOMs is 45, a value approximately 1.9-fold higher compared to that of EDLs. This higher content of Stim1 and Orai1 supports the idea that SOCE is a robust component of calcium signaling in EOMs mediating the constant calcium entry necessary to replenish sarcoplasmic reticulum stores needed to support the continuous fast muscle contraction unique to EOMs, compared to other striated muscles. A mind-boggling result emerging from the quantitative analysis of Stim1 and Orai1 is that slow twitch muscles such a soleus contain very low levels of Orai1 protein which could not be quantified by LC-MS. This raises the important question as to the nature of the molecular component(s) interacting with Stim1 in order to operate SOCE in slow twitch muscles. At this point in time, we cannot exclude the possibility that in slow twitch muscles, Stim1 interacts with a molecular partner different from Orai1, or that SOCE might be operated by an Orai1 variant having a much higher divalent cation conductance compared to the 'classical' Orai1 isoform expressed in EDL and EOM. Nevertheless, such a question is beyond the scope of the present investigation and cannot be answered by the data presented here.

*STAC3* mutations have been linked to Native American Myopathy (NAM), a severe congenital myopathy resulting in muscle weakness and skeleton alteration (*Horstick et al., 2013*). Such mutations cause a decrease of the interaction between Stac3 with DHPRα1s resulting in a functional deficit of EC coupling (*Wong King Yuen et al., 2017*). On the basis the quantitative data we obtained using the LC-MS standard curve generated by spiked-in peptides, the Stac3 to Cacna1s stoichiometry ratio is 1.11, 1.22, and 1.67 in EDL, soleus and EOM respectively, and no differences were observed between WT and dHT mice. Stac3 interacts via its SH3 domain with a Kd ranging between 2 and 10 μM, with

**Table 3.** Relative change in the content of selected proteins in EOM isolated from WT and dHT mice.

| | Gene name | Protein* | Relative content | q value |
|---|---|---|---|---|
| | *Ryr1* | Ryanodine receptor 1 (RyR1) | 0.42 | $1.73 \times 10^{-6}$ |
| | *Asph* | Aspartyl/asparaginyl ß-hydroxylase (junctin/junctate/aspß-hydroxylase) | 1.35 | 0.00028 |
| | *Casq2* | Calsequestrin-2 | 1.45 | 0.00031 |
| | *Casq1* | Calsequestrin-1 | 1.55 | 0.0063 |
| ECC | *ATP2a2* | Sarcoplasmic/endoplasmic reticulum calcium ATPase 2 (SERCA2) | 1.55 | 0.00052 |
| | *Myh13* | MyHC-EO | 0.27 | $1.01 \times 10^{-5}$ |
| | *Actn2* | α -actinin 2 | 1.36 | 0.0047 |
| | *Myh7b* | Myosin-7B (MyH7B, cardiac musle ß isoform, MyHC14) | 1.44 | 0.0038 |
| | *Actn1* | α -actinin 1 | 1.45 | 0.015 |
| | *Tnnt2* | Troponin T, cardiac isoform | 1.45 | 0.0037 |
| | *Myot* | Myotilin | 1.61 | 0.0018 |
| | *Tnnt1* | Troponin T slow, skeletal muscle (TnTs) | 1.85 | 0.022 |
| | *Myoz3* | Myozenin 3 | 2.03 | $1.29 \times 10^{-5}$ |
| | *Myh6* | Myosin 6 (MyHC cardiac muscle α-isoform) | 2.22 | $1.01 \times 10^{-5}$ |
| Contractile proteins | *Tnnc1* | Troponin C, slow skeletal and cardiac (TN-C) | 2.61 | $8.1 \times 10^{-5}$ |
| | *Itga7* | Integrin α 7 | 1.29 | 0.000191 |
| | *Col6a5* | Collagen (VI) α –5 chain | 1.32 | 0.0033 |
| | *Col6a6* | Collagen (VI) α –6 chain | 1.32 | 0.0087 |
| | *Col12a1* | Collagen (XII) α –1 chain | 1.34 | 0.012 |
| | *Col14a1* | Collagen (XIV) α –1 chain | 1.61 | 0.00072 |
| Collagen and ECM proteins | *Col11a2* | Collagen (XI) α –2 chain | 4.46 | $1.27 \times 10^{-5}$ |
| Heat shock proteins | *Hspa9* | Mitochondrial, stress-70 protein | 0.78 | 0.013 |
| | *Hsp90b1* | Hsp 90b1 (GRP-94; 90 kDa glucose regulated protein) | 1.26 | 0.0052 |
| | *Hspb3* | Hsp ß- 3 | 1.27 | 0.0056 |
| | *Dnaja1* | Dnaj homolog subfamily A member 1 (Hsp 40 kDa protein 4) | 1.28 | 0.0067 |
| | *Hspb1* | Hsp ß–1 (Hsp25) | 1.29 | 0.00029 |
| | *Hspa5* | Heat Shock Protein Family A (Hsp70) Member 5 (BiP) | 1.32 | 0.00033 |
| | *Hspa1a* | Heat shock 70 kDa protein 1 A | 1.33 | 0.0069 |
| | *Hsp90aa1* | Hsp 90 a | 1.40 | 0.00094 |
| | *Dnajb1* | Dnaj homolog subfamily B member 1 (Hsp40) | 1.46 | 0.00021 |
| | *Dnajb4* | DnaJ homolog subfamily B member 4 (Hsp40) | 1.54 | 0.011 |
| | *Hspb6* | Hsp ß- 6 | 1.61 | $4.05 \times 10^{-5}$ |

*Table 3 continued on next page*

*Table 3 continued*

| | Gene name | Protein* | Relative content | q value |
|---|---|---|---|---|
| | *Rps2* | 40 S ribosomal protein S2 | 1.260 | 0.004 |
| | *Rpsa* | 40 S ribosomal protein SA | 1.276 | 0.0035 |
| | *Rps11* | 40 S ribosomal protein S11 | 1.321 | 0.0016 |
| | *Rps20* | 40 S ribosomal protein S20 | 1.324 | 0.0007 |
| | *Rpl10* | 60 S ribosomal protein L10 | 1.332 | 0.0035 |
| | *Rplp2* | 60 S acidic ribosomal protein P2 | 1.332 | 0.00048 |
| | *Rpl11* | 60 S ribosomal protein L11 | 1.335 | 0.0077 |
| | *Rps28* | 40 S ribosomal protein S28 | 1.346 | 0.0041 |
| | *Rpl3* | 60 S ribosomal protein L13 | 1.385 | $1.42 \times 10^{-5}$ |
| | *Rps7* | 40 S ribosomal protein S7 | 1.396 | 0.0125 |
| | *Rpl27a* | 60 S ribosomal protein L27a | 1.461 | 0.00034 |
| Ribosomal Proteins | *Rps27a* | 40 S ribosomal protein S27a | 1.570 | 0.038 |
| FK506 binding proteins | *Fkbp1a* | Peptidyl-prolyl cis-trans isomerase FKBP1A (FKBP12; calstabin-1) | 0.79 | 0.015 |
| Calcium-dependent protein kinases | *Camk2b* | Calcium/Calmodulin-Dependent Protein Kinase IIß | 1.30 | 0.0018 |
| | *Camk2d* | Calcium/Calmodulin-Dependent Protein Kinase IIδ | 2.32 | $8.35 \times 10^{-6}$ |
| | *S100a16* | S100 A16 | 1.29 | 0.014 |
| Calcium binding proteins | *S100a1* | S100 A1 | 1.30 | 0.029 |

*The nomenclature of Proteins is based on that of the UniProtKB database.

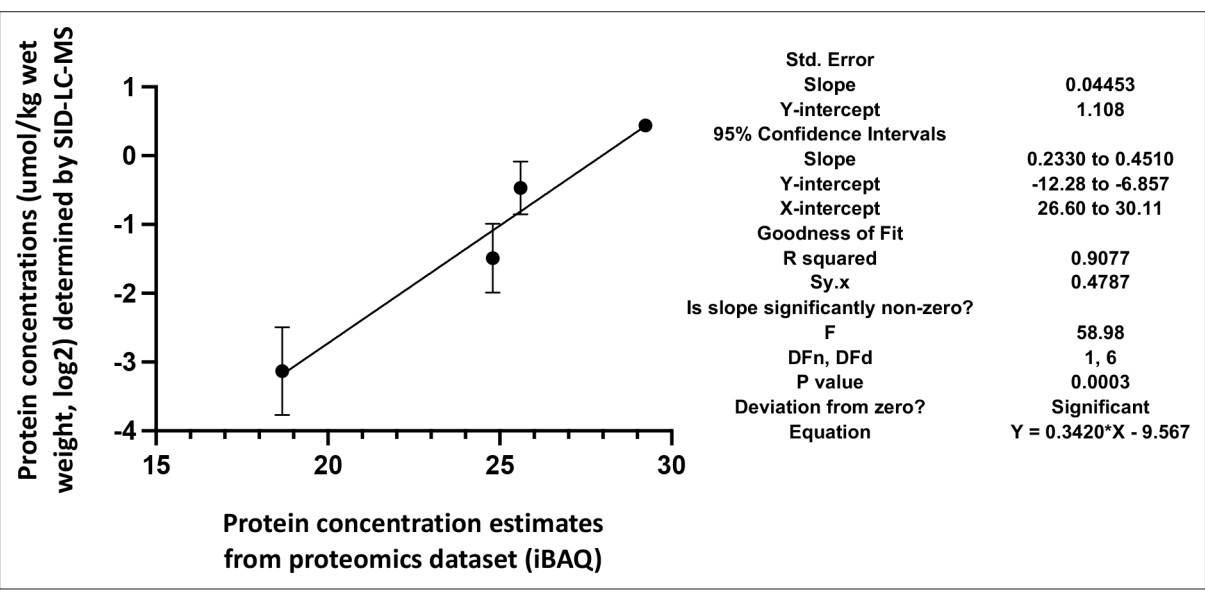

**Figure 5.** Correlation of the actual cellular abundances of four selected proteins (in μmol/kg wet weight) determined by PRM/SID (n=2) and the iBAQ values (n=5) determined by label-free/TMT quantification (both in logarithmic scale, base 2) from the global proteomics discovery dataset for EDL samples. Error bars are indicated for the y-axis, but for the x-axis, due to their low scale (range from 0.058 to 0.086), they are not shown by the software PRISM, GraphPad Software, (v9). The simple linear regression results obtained by PRISM GraphPad Software, (v9) are shown on the right.

**Table 4.** Concentration µmol/Kg (mean ± SD) of proteins involved in ECC in EDL, soleus and EOM muscles from WT (n=5 mice) and dHT (n=5 mice) using the peptide 4 point calibration curve.

| Gene name | EDL | | soleus | | EOM | |
|---|---|---|---|---|---|---|
| | WT | dHT | WT | dHT | WT | dHT |
| *Ryr1* monomers (terameric channel) | 1.29±0.07 (0.32) | 0.86±0.01 (0.21) | 0.49±0.02 (0.12) | 0.40±0.002 (0.10) | 0.59±0.02 (0.15) | 0.35±0.01 (0.09) |
| *Cacna1s* | 0.56±0.03 | 0.49±0.01 | 0.18±0.01 | 0.16±0.002 | 0.21±0.01 | 0.19±0.004 |
| *Stac3* | 0.62±0.07 | 0.53±0.06 | 0.22±0.02 | 0.20±0.01 | 0.17±0.01 | 0.15±0.01 |
| *Jsrp1* | 0.42±0.03 | 0.40±0.01 | 0.32±0.01 | 0.29±0.03 | 0.35±0.01 | 0.35±0.02 |
| *Asph* | 0.21±0.01 | 0.26±0.01 | 0.30±0.02 | 0.35±0.03 | 0.82±0.03 | 1.00±0.03 |
| *Trdn* | 0.96±0.18 | 0.79±0.06 | 0.16±0.03 | 0.13±0.01 | 0.23±0.01 | 0.22±0.01 |
| *Jph1* | 0.71±0.09 | 0.58±0.04 | 0.29±0.02 | 0.25±0.01 | 0.24±0.01 | 0.23±0.01 |
| *Stim1* | 0.46±0.02 | 0.48±0.03 | 0.55±0.03 | 0.56±0.03 | 1.35±0.03 | 1.42±0.09 |
| *Orai1* monomers (6-subunt complex) | 0.11±0.01 (0.02) | 0.13±0.02 (0.02) | Not detected | Not detected | 0.16±0.03 (0.03) | 0.17±0.01 (0.03) |

a binding site within the cytosolic II-III loop of the DHPRα1s (*Rufenach et al., 2020*; *Polster et al., 2018*). Here we show that the molar content of Stac3 in EDL, soleus and EOMs is between 2.5- and 10-fold lower than its Kd for the DHPRα1s II-III loop binding site (*Wong King Yuen et al., 2017*; *Rufenach et al., 2020*). Thus, the fractional occupancy of the DHPRα1s binding site by Stac3 is lower than 50%, a value which is still sufficient to support normal EC coupling. Nevertheless, the extent of the fractional occupancy depends on the fiber type. In particular, if the Kd of the DHPRα1s binding site for Stac3 is identical in EDL, soleus and EOM, then the fractional occupancy of the DHPRα1s binding site for Stac3 in soleus and EOMs is lower than of EDL muscles, because the molar content of Stac3 in soleus and EOMs is three-fold lower compared to that of EDL (*Table 4*). *STAC3* mutations linked to NAM decrease the Kd of the SH3 domain of Stac3 for the cytosolic II-III loop of DHPRα1s (*Rufenach et al., 2020*) further lowering the fractional occupancy of Stac3 binding site of DHPRα1s to a low level close to zero, a condition that would disrupt EC coupling in NAM patients (*Horstick et al., 2013*).

## Recessive *Ryr1* mutations affect the expression of extracellular matrix

'Reactome' interaction pathway analysis revealed that the major pathways affected by the presence of compound heterozygous *Ryr1* mutations in EDL muscles includes proteins involved in organization and degradation of the extracellular matrix (ECM) and indeed the content of collagen I, II, IV, V, and XI was significantly reduced. The ECM plays an important role in muscle force transmission, maintenance and repair and collagen fibers account for 1–10% muscle dry weight, forming a highly ordered network surrounding individual muscle fibers and muscle bundles (*Trotter and Purslow, 1992*; *Gillies and Lieber, 2011*). Exactly how defects in the collagen network impact muscle function is not clear, nevertheless patients bearing mutations in Collagen VI (*COL6A1*, *COL6A2* and *COL6A3*) suffer from Ulrich and Bethlem myopathies (*Bethlem and Wijngaarden, 1976*) and exhibit muscle contractures involving elbows and ankles, a clinical sign that has been also described in patients suffering of congenital myopathies linked to recessive *RYR1* mutations (*Monnier et al., 2008*; *Klein*

**Table 5.** Calculated ratio values.

| Gene name | EDL | | soleus | | EOM | |
|---|---|---|---|---|---|---|
| | WT | dHT | WT | dHT | WT | dHT |
| *Ryr1* complex/*Cacna1s* | 0.571 | 0.429 | 0.667 | 0.625 | 0.714 | 0.474 |
| *Stac3/Cacna1s* | 1.11 | 1.08 | 1.22 | 1.25 | 1.67 | 1.84 |
| *Jsrp1/Cacna1s* | 0.75 | 0.82 | 1.78 | 1.81 | 0.95 | 1.00 |
| *Stim1/Orai1* complex | 23.0 | 24.0 | - | - | 45.0 | 47.3 |

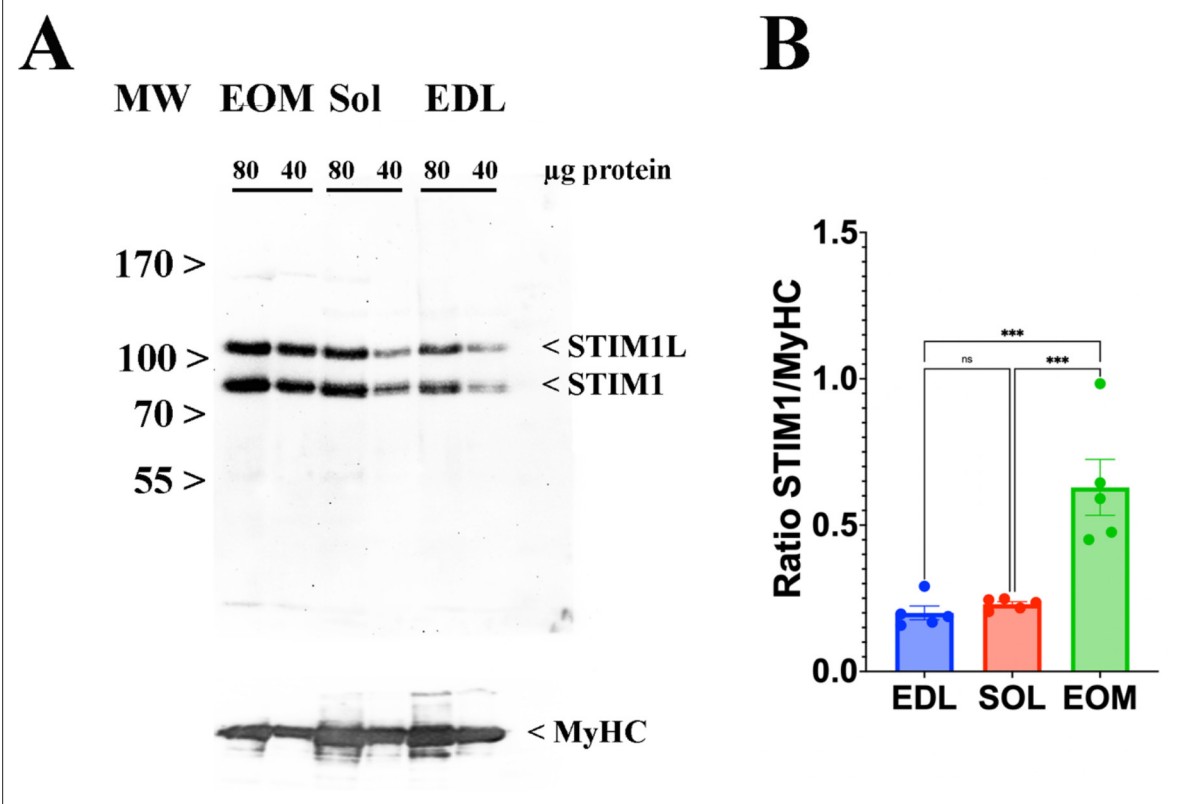

**Figure 6.** EOMs are enriched in Stim 1. (**A**) Representative western blots showing Stim1 and Stim1L immunopositive bands. Forty and eighty micrograms of total homogenates from EOM, soleus, and EDL muscles isolated from WT mice were loaded onto a 7.5% SDS PAGE. Proteins were blotted onto nitrocellulose, probed with an antibody recognizing Stim1 and Stim1L, followed by incubation with an anti-rabbit IgG HRP-linked antibody. Bands were visualized by chemiluminescence. Blots were subsequently stripped and probed with anti-MyHC (all) for loading normalization (bottom panel). (**B**) Relative content of Stim1 in the three muscle types examined. Each symbol represents the value of a single mouse. *** p<0.001.

The online version of this article includes the following source data for figure 6:

**Source data 1.** Bots refer to data shown in *Figure 6A*.

**Source data 2.** Bots refer to data used for statistical analysis depicted in *Figure 6B*.

*et al., 2012*). In addition, a frequent common feature of patients with congenital myopathies carrying recessive *RYR1* mutations is the appearance of a number of skeletal abnormalities at birth, including scoliosis and congenital dislocation of the hip, kyphosis, clubfoot, flattening of the arch of the foot (or an abnormally high arch of the foot). Patients also exhibit joint laxity that may lead to dislocation of the patella, or, more rarely abnormal tightening of certain joints, resulting in contractures especially of the Achilles tendon (*Jungbluth, 2007*) and such features may also be related to the changes in collagen composition of the muscle fibers, as observed in dHT mice.

## Recessive *Ryr1* mutations affect the expression of proteins involved in ER protein quality control and protein synthesis

Muscles from the dHT did not show upregulation of proteins related to ER stress such as PERK, IRE1a, ATF6, although the content of BiP as well as that of several heat shock proteins was significantly increased in EDL and EOM muscles. Heat shock proteins are molecular chaperones that participate in the safeguard of cell integrity, playing numerous functions including protection from heat insults, prevention of aggregation and facilitation of protein folding. There are different categories of heat shock proteins, including small HSPs (HSPB1-10) that are involved in protein folding, prevention of aggregation. In skeletal muscle, these small proteins have been shown to be involved in the maintenance of the cytoskeletal network and contractile elements and play a role in myogenic differentiation (*Dreiza et al., 2010*; *Maglara et al., 2003*; *Martin et al., 1997*). Large HSP are present in many subcellular locations including mitochondria, nucleus, sarcoplasmic reticulum and myoplasm

where they facilitate protein folding and re-folding, facilitate protein transport into the SR and mitochondria and prevent aggregate formation. Interestingly, intensive resistance training increases heat shock protein levels in muscle (**Murlasits et al., 2006**) whereas aging is associated with a decrease in HSP70 response in muscles following muscle contraction (**Vasilaki et al., 2002**; **Murgia et al., 2017**; **Brinkmeier and Ohlendieck, 2014**). Proteomic profiling has shown that muscle diseases including dysferlinopathies, myofibrillar myopathies, spinal muscular atrophy, Duchenne muscular dystrophy and others are associated with upregulation of distinct HSP (for review see 51). These results together with our findings suggest that altered muscle function caused by genetic mutations are accompanied by adaptive cellular responses aimed at counterbalancing muscle damage and/or restoring proper function. Of note, in soleus muscles from the dHT mice, HSP are not up-regulated; this may be due to the fact that soleus muscles are less damaged/stressed or because the content of HSPs in soleus muscles is constitutively higher than in EDL muscles. In particular, the expression of HSP70 is fivefold higher in WT soleus compared to WT EDL muscles. The high expression of HSP70 might thus protect slow twitch muscles from extensive damage linked to the expression of mutant RyR1s, an event which may ultimately account for the fiber type I predominance observed in patients with congenital myopathies linked to *RYR1* mutations.

An interesting observation of the present study is that the content of ribosomal proteins constituting the 40 S and 60 S subunits is significantly increased in the three muscle types from the dHT compared to WT. In skeletal muscle, up-regulation of ribosomal proteins accompanies hypertrophy and training, whereas ribosomal proteins decrease with age (**Ubadia-Mohien et al., 2019**). Thus, our results indicate that the presence of *Ryr1* mutations evoke a global adaptive response aimed at (i) preserving the integrity of intracellular protein compartments and (ii) increasing muscle protein turnover.

## Differentially expressed proteins shared by EDL, soleus, and EOM from dHT mice

We postulated that some of the skeletal muscle phenotypic features of dHT might be linked to changes in protein expression shared by the three muscles types we analyzed. We found that RyR1 is the only shared downregulated protein, while 39 proteins are upregulated, including calmodulin kinase 2δ, aspartyl-ß-hydroxylase and Kelch like protein 41. Mutations in the Kelch like protein 41 (*KLH41*) have been identified in patients with nemaline myopathy (**Gupta et al., 2013**) and it has been suggested that this Kelch like protein regulates muscle protein homeostasis through its interaction with the muscle-specific E3 ubiquitin ligase MuRF 1. Furthermore, *KLH41* appears to be activated in response to muscle damage suggesting that it may play a role in muscle adaptation and repair (**Gupta et al., 2013**).

## Conclusions

Multiplexed proteomic analysis is a powerful approach for the quantitative proteomic analysis of a variety of biological samples. In particular, absolute quantification can be achieved by measuring the content of a protein relative to a spiked-in peptide with known absolute concentration. A limitation of the multiplexed isobaric mass tag-based protein quantification is the reliable detection of very low abundant proteins, such as transcriptional factors and other molecules involved in cellular signaling. Because of this intrinsic hurdle of multiplex isobaric mass tag spectrometry, in this study we missed nuclear proteins in addition to protein components of signaling pathways. An additional drawback of this study is that it gives a static image of muscle protein content in young adult mice without conveying information about the dynamics of protein changes or changes in post-translational modifications occurring during muscle disease.

In summary, our quantitative proteomic study, shows that recessive *Ryr1* mutations not only decrease the content of RyR1 protein in muscle, but also affect the content of many other proteins involved in a variety of biological processes.

# Materials and methods

## Compliance with ethical standards

All experiments involving animals were carried out on 12 weeks old male WT and dHT mice littermates. Experimental procedures were approved by the Cantonal Veterinary Authority of Basel Stadt (BS Kantonales Veterinäramt Permit numbers 1728). All experiments were performed in accordance with relevant guidelines and regulations.

## Proteomics analysis using tandem mass tags

EDL, soleus and EOM muscles from 5 male WT and 5 male dHT, 12 weeks old mice were excised, weighed, snap frozen in liquid nitrogen and mechanically grinded. Approximately 10 mg of EDL, 8 mg for of Soleus and 6 mg of EOM muscle tissue was grinded and subsequently lysed in 200 µl of lysis buffer containing 100 mM TRIS, 1% sodium deoxycholate (SDC), 10 mM TCEP and 15 mM chloroacetamide, followed by sonication (Bioruptor, 20 cycles, 30 s on/off, Diagenode, Belgium) and heating to 95 °C for 10 min. After cooling, protein samples were digested by incubated overnight at 37 °C with sequencing-grade modified trypsin (1/50, w/w; Promega, Madison, Wisconsin). Samples were acidified using 5% TFA and peptides cleaned up using the Phoenix 96 x kit (PreOmics, Martinsried, Germany) following the manufacturer's instructions. After drying the peptides in a SpeedVac, samples were stored at –80 °C.

Dried peptides were dissolved in 100 µl of 0.1% formic acid and the peptide concentration determined by UV-nanodrop analysis. Sample aliquots containing 25 µg of peptides were dried and labeled with tandem mass isobaric tags (TMT 10-plex, Thermo Fisher Scientific) according to the manufacturer's instructions. To control for ratio distortion during quantification, a peptide calibration mixture consisting of six digested standard proteins mixed in different amounts were added to each sample before TMT labeling as recently described (*Ahrné et al., 2016*). After pooling the differentially TMT-labeled peptide samples, peptides were again desalted on C18 reversed-phase spin columns according to the manufacturer's instructions (Macrospin, Harvard Apparatus) and dried under vacuum. Half of the pooled TMT-labeled peptides (125 µg of peptides) were fractionated by high-pH reversed phase separation using a XBridge Peptide BEH C18 column (3,5 µm, 130 Å, 1 mm x 150 mm, Waters) on an Agilent 1260 Infinity HPLC system. 125 µg of peptides were loaded onto the column in buffer A (ammonium formate [20 mM, pH 10, in water]) and eluted using a two-step linear gradient starting from 2% to 10% in 5 min and then to 50% (v/v) buffer B 90% acetonitrile / 10% ammonium formate (20 mM, pH 10) over 55 min at a flow rate of 42 µl/min. Elution of peptides was monitored with a UV detector (215 nm, 254 nm). A total of 36 fractions were collected, pooled into 12 fractions using a post-concatenation strategy as previously described (*Obradović et al., 2019*; *Kulyyassov et al., 2021*) and dried under vacuum.

The generated 12 peptide samples fractions were analyzed by LC-MS as described previously (*Ahrné et al., 2016*). Chromatographic separation of peptides was carried out using an EASY nano-LC 1000 system (Thermo Fisher Scientific), equipped with a heated RP-HPLC column (75 µm x 37 cm) packed in-house with 1.9 µm C18 resin (Reprosil-AQ Pur, Dr. Maisch). Aliquots of 1 µg of total peptides of each fraction were analyzed per LC-MS/MS run using a linear gradient ranging from 95% solvent A (0.15% formic acid, 2% acetonitrile) and 5% solvent B (98% acetonitrile, 2% water, 0.15% formic acid) to 30% solvent B over 90 min at a flow rate of 200 nl/min. Mass spectrometry analysis was performed on Q-Exactive HF mass spectrometer equipped with a nanoelectrospray ion source (both Thermo Fisher Scientific). Each MS1 scan was followed by high-collision-dissociation (HCD) of the 10 most abundant precursor ions with dynamic exclusion for 20 s. Total cycle time was approximately 1 s. For MS1, 3e6 ions were accumulated in the Orbitrap cell over a maximum time of 100ms and scanned at a resolution of 120,000 FWHM (at 200 m/z). MS2 scans were acquired at a target setting of 1e5 ions, accumulation time of 100ms and a resolution of 30,000 FWHM (at 200 m/z). Singly charged ions and ions with unassigned charge state were excluded from triggering MS2 events. The normalized collision energy was set to 35%, the mass isolation window was set to 1.1 m/z and one microscan was acquired for each spectrum.

The acquired raw-files were searched against a protein database containing sequences of the predicted SwissProt entries of *Mus musculus* (https://www.ebi.ac.uk/, release date 2019/03/27), *Myh2* and *Myh13* from Trembl, the six calibration mix proteins (*Ahrné et al., 2016*) and commonly observed contaminants (in total 17,414 sequences) using the SpectroMine software (Biognosys, version

1.0.20235.13.16424) and the TMT 10-plex default settings. In brief, the precursor ion tolerance was set to 10 ppm and fragment ion tolerance was set to 0.02 Da. The search criteria were set as follows: full tryptic specificity was required (cleavage after lysine or arginine residues unless followed by proline), 3 missed cleavages were allowed, carbamidomethylation (C), TMT6plex (K and peptide n-terminus) were set as fixed modification and oxidation (M) as a variable modification. The false identification rate was set to 1% by the software based on the number of decoy hits. Proteins that contained similar peptides and could not be differentiated based on MS/MS analysis alone were grouped to satisfy the principles of parsimony. Proteins sharing significant peptide evidence were grouped into clusters. Acquired reporter ion intensities in the experiments were employed for automated quantification and statistically analyzed using a modified version of our in-house developed SafeQuant R script (v2.3)(*Ahrné et al., 2016*). This analysis included adjustment of reporter ion intensities, global data normalization by equalizing the total reporter ion intensity across all channels, summation of reporter ion intensities per protein and channel, calculation of protein abundance ratios and testing for differential abundance using empirical Bayes moderated t-statistics. Finally, the calculated p-values were corrected for multiple testing using the Benjamini−Hochberg method.

## Targeted PRM-LC-MS analysis of RyR1 and DHPRα1s, Stim1 and Orai1

In a first step, parallel reaction-monitoring (PRM) assays (*Kulyyassov et al., 2021*) were generated from a mixture containing 50 fmol of each proteotypic heavy reference peptide of the target proteins (AIWAEYDPEAK, GEGIPTTAK, TGGLFGQVDNFLER (for DHPRα1s); AGDVQSGGSDQER, GPHLVGPSR, SNQDLITENLLPGR, TLLWTFIK, VVAEEEQLR (for Ryr1); LISVEDLWK, AIDTVIFGP-PIITR, ITEPQIGIGSQR, LSFEAVR, YAEEEIEQVR (for Stim1); QFQELNELAEFAR, IQDQIDHR, SLVSHK (for Orai1); JPT Peptide Technologies GmbH) plus iRT peptides (Biognosys, Schlieren, Switzerland). Peptides were subjected to LC–MS/MS analysis using a Q Exactive Plus mass spectrometer fitted with an EASY-nLC 1000 (both Thermo Fisher Scientific) and a custom-made column heater set to 60 °C. Peptides were resolved using a RP-HPLC column (75 μm×30 cm) packed in-house with C18 resin (ReproSil-Pur C18–AQ, 1.9 μm resin; Dr. Maisch GmbH) at a flow rate of 0.2 μLmin-1. A linear gradient ranging from 5% buffer B to 45% buffer B over 60 min was used for peptide separation. Buffer A was 0.1% formic acid in water and buffer B was 80% acetonitrile, 0.1% formic acid in water. The mass spectrometer was operated in DDA mode with a total cycle time of approximately 1 s. Each MS1 scan was followed by high-collision-dissociation (HCD) of the 20 most abundant precursor ions with dynamic exclusion set to 5 seconds. For MS1, 3e6 ions were accumulated in the Orbitrap over a maximum time of 254ms and scanned at a resolution of 70,000 FWHM (at 200 m/z). MS2 scans were acquired at a target setting of 1e5 ions, maximum accumulation time of 110ms and a resolution of 35,000 FWHM (at 200 m/z). Singly charged ions, ions with charge state ≥6 and ions with unassigned charge state were excluded from triggering MS2 events. The normalized collision energy was set to 27%, the mass isolation window was set to 1.4 m/z and one microscan was acquired for each spectrum. The acquired raw-files were searched using the MaxQuant software (Version 1.6.2.3) against the same protein sequence database as described above using default parameters except protein, peptide and site FDR were set to 1 and Lys8 and Arg10 were added as variable modifications. The best 6 transitions for each peptide were selected automatically using an in-house software tool and imported into SpectroDive (version 8, Biognosys, Schlieren). A scheduled (window width 12 min) mass isolation list containing the iRT peptides was exported form SpectroDive and imported into the Q Exactive plus operating software for PRM analysis.

Peptide samples for PRM analysis were resuspended in 0.1% aqueous formic acid, spiked with iRT peptides and the heavy reference peptide mix at a concentration of 10 fmol of heavy reference peptides per 1 μg of total endogenous peptide mass and subjected to LC–MS/MS analysis on the same LC-MS system described above using the following settings: The resolution of the orbitrap was set to 140,000 FWHM (at 200 m/z), the fill time was set to 500ms to reach an AGC target of 3e6, the normalized collision energy was set to 27%, ion isolation window was set to 0.4 m/z and the first mass was fixed to 100 m/z. A MS1 scan at 35,000 resolution (FWHM at 200 m/z), AGC target 3e6 and fill time of 50ms was included in each MS cycle. All raw-files were imported into SpectroDive for protein / peptide quantification. To control for variation in injected sample amounts, the total ion chromatogram (only comprising ions with two to five charges) of each sample was determined and used for normalization. To this end, the generated raw files were imported into the Progenesis QI software

(Nonlinear Dynamics (Waters), Version 2.0), the intensity of all precursor ions with a charge of +2 to+5 were extracted, summed for each sample and used for normalization. Normalized ratios were transformed from the linear to the log-scale, normalized relative to the control condition and the median ratio among peptides corresponding to one protein was used for protein quantification.

## Western blot analysis of Stim1 and Stim1L

Total homogenates of EDL, soleus and EOM muscles from WT mice were prepared in cracking buffer as previously described (*Elbaz et al., 2019*; *Eckhardt et al., 2020*). Proteins were separated on a 7.5% SDS-PAG, blotted onto nitrocellulose and probed with an antibody recognizing Stim1 and Stim1L (1/2000 anti-STIM1, Millipore, #AB9870), followed by incubation with an anti-rabbit IgG HRP-linked antibody (1/6000, Cell Signaling Technology, #7074). Bands were visualized by chemiluminescence. Blots were subsequently stripped and probed with anti-MyHC (all) (1/5000, DSHB, #MF20) for loading normalization as previously described (*Elbaz et al., 2019*; *Eckhardt et al., 2020*). Statistical analysis was performed using a one-way ANOVA test.

## Data-analyses

Matlab 2021b (Mathworks) (*Darik, 2022*) was used to process the proteomics data and to generate heatmap, volcano plots and Venn diagrams. 'Reactome' is based on 'multiple proteins' or 'Proteins with Values/Ranks' string analyses (https://string-db.org/; Version 11.5).

## Acknowledgements

The authors wish to acknowledge Dr. Eric Ahrné for constructive discussions.

## Additional information

### Funding

| Funder | Grant reference number | Author |
| --- | --- | --- |
| Swiss National Science Foundation | 31003A-184765 | Susan Treves |
| NeRab | | Susan Treves |
| RYR-1 Foundation | | Francesco Zorzato |
| Swiss Foundation for Research on Muscle Diseases | | Susan Treves Francesco Zorzato |
| Swiss National Science Foundation | 310030-212192 | Susan Treves |

The funders had no role in study design, data collection and interpretation, or the decision to submit the work for publication.

### Author contributions

Jan Eckhardt, Data curation, Formal analysis, Validation, Investigation, Writing – review and editing; Alexis Ruiz, Data curation, Investigation, Methodology, Writing – review and editing; Stéphane Koenig, Formal analysis, Investigation, Writing – review and editing; Maud Frieden, Investigation, Methodology, Writing – review and editing; Hervé Meier, Data curation, Software, Formal analysis, Investigation, Methodology, Writing – review and editing; Alexander Schmidt, Conceptualization, Resources, Data curation, Software, Formal analysis, Supervision, Funding acquisition, Writing – original draft, Project administration, Writing – review and editing; Susan Treves, Conceptualization, Resources, Data curation, Formal analysis, Supervision, Funding acquisition, Validation, Investigation, Writing – original draft, Project administration, Writing – review and editing; Francesco Zorzato, Conceptualization, Data curation, Formal analysis, Supervision, Funding acquisition, Validation, Investigation, Methodology, Writing – original draft, Project administration, Writing – review and editing

## Author ORCIDs
Maud Frieden http://orcid.org/0000-0001-7135-0874
Susan Treves http://orcid.org/0000-0002-0007-9631
Francesco Zorzato http://orcid.org/0000-0002-8469-7065

## Ethics

All experiments involving animals were carried out on 12 weeks old male wild type and dHT mice littermates. Experimental procedures were approved by the Cantonal Veterinary Authority of Basel Stadt (BS Kantonales Veterinäramt Permit number 1728). All experiments were performed in accordance with relevant guidelines and regulations.

## Decision letter and Author response
Decision letter https://doi.org/10.7554/eLife.83618.sa1
Author response https://doi.org/10.7554/eLife.83618.sa2

---

# Additional files

### Supplementary files
• Supplementary file 1. Tables of relative changes in protein content between soleus and EDL, EDL and EOM and soleus and EOM muscles iin WT mice. (a) Table showing relative change in protein content between soleus and EDL muscles isolated from WT mice. (b) Table showing relative change in protein content between EOM and EDL muscles isolated from WT mice. (c) Table showing relative change in protein content between EOM and soleus muscles isolated from WT mice.

• MDAR checklist

### Data availability
All data, code, and materials used in the analysis are available in some form to any researcher for purposes of reproducing or extending the analysis. Mass spectrometry data has been deposited to the ProteomeXchange Consortium via the Pride partner repository (http://ebi.ac.uk/pride) with the following access number PXD036789 (http://www.ebi.ac.uk/pride/archive/projects/PXD036789). Original western blot figures have been uploaded as a zipped file as source data 1 and 2.

The following dataset was generated:

| Author(s) | Year | Dataset title | Dataset URL | Database and Identifier |
|---|---|---|---|---|
| Schmidt A | 2022 | Proteomic analysis of skeletal muscles from wild type and transgenic mice carrying recessive Ryr1 mutations | http://www.ebi.ac.uk/pride/archive/projects/PXD036789 | PRIDE, PXD036789 |

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
