## [Editor Report]

This is a fundamental study reporting a comprehensive proteomic analysis in three skeletal muscle types from wild-type and RYR1-related myopathy mice. It adds quantitative stoichiometry of several excitation-contraction coupling-related proteins. This valuable work compares the disease-related proteomes of the different skeletal muscle groups.

---

## [Decision Letter]

**Decision letter after peer review:**

Thank you for submitting your article "Quantitative proteomic analysis of skeletal muscles from wild type and transgenic mice carrying recessive *Ryr1* mutations linked to congenital myopathies" for consideration by *eLife*. Your article has been reviewed by 3 peer reviewers, one of whom is a member of our Board of Reviewing Editors, and the evaluation has been overseen by Mone Zaidi as the Senior Editor.

The reviewers have discussed their reviews with one another, and the Reviewing Editor has drafted this to help you prepare a revised submission. Reviewers (1) and (2) were enthusiastic about the work; reviewer (3) had reservations. Nevertheless we would like to offer you the opportunity to submit a revised version addressing the comments particularly of reviewers (2) and (3). I have worked through all the comments and hope the notes to follow may be useful in connection with this.

Essential revisions:

Reviewer (2) makes the following broad comments that may merit addressing:

a) The main limitation of this study is that the results are primarily descriptive in nature, and thus, do not provide mechanistic insight into how Ryr1 disease mutations lead to the muscle-specific changes observed in the EDL, soleus and EOM proteomes.

b) Results comparing fast twitch (EDL) and slow twitch (soleus) muscles from WT mice confirmed several known differences between the two muscle types. Similar analyses between EOM/EDL and EOM/soleus muscles from WT mice were not conducted.

c) While a reactome pathway analysis for proteins changes observed in EDL is shown in Supplemental Figure 1, the authors do not fully discuss the nature of the proteins and corresponding pathways impacted in the other two muscle groups analyzed.

Specific requests

Reviewer (2) also makes the following specific requests:

a) This study would be strengthened by inclusion of tables that summarize relative protein changes between EOM/EDL muscles and EOM/soleus muscles from WT mice as was done in Table 1 for soleus/EDL muscles from WT mice.

b) As there are reliable antibodies for many of the selected proteins found to be significantly increased or decreased in muscle from dHT mice, it would seem important and feasible to validate at least a subset of the proteomic findings in western blot experiments. For example, the proteome studies found JP-1 levels to be reduced in both EDL and soleus, but not in EOM; Hspa5 (BIP) levels to be increased in EDL and EOM, but unaltered in soleus; collagen levels to be reduced in EDL, but unaltered in both soleus and EOM. Such findings could readily be validated in quantitative western blot experiments.

c) Beyond the author-selected muscle-related processes shown in Tables 2-4, limited non-biased information is provided regarding other potential proteins and pathways altered in the different muscles evaluated from dHT mice. A more robust unbiased evaluation of all of the significantly increased/decreased proteins and the underlying pathways involved could be informative. A reactome pathway analysis comparison is shown for EDL muscles from WT and dHT mice in Supplemental Figure 1. It would be helpful if the authors could include similar reactome pathway analyses for the proteins found to be significantly increased/decreased in soleus and EOM of WT and dHT mice.

d) The authors provide a Venn diagram in Supplemental Figure 2 to highlight overlap between proteins that are increased and decreased across all three muscles. Interestingly, for all three muscles evaluated (EDL, soleus, EOM), more proteins were found to be increased than decreased in dHT mice (e.g. 560 proteins are increased in EOM and only 117 are decreased). In addition, while only one protein (Ryr1) was found to be decreased across all muscles, 40 proteins were found to be increased in all three muscles. It would be helpful if the authors could provide a summary table that lists these 40 proteins that are upregulated in all three muscles. Does a GO pathway analysis of these 40 proteins provide any common theme that extends across all three types muscles? Are there any common themes to come out of separate GO pathway analyses of the 848 protein differences observed in EDL, 509 protein differences observed in soleus, and 677 protein differences observed in EOM? What about the union of common proteins changes observed between EDL and EOM (144 proteins), EDL and soleus (98 proteins), and EOM and soleus (106 proteins)?

Reviewer (3) Broad comments:

a) it would be useful to determine whether changes in protein levels correlated with changes in mRNA levels and whether or not the protein present was functional, and whether Stac3 was in fact stoichiometrically depleted in relation to Cacna1s.

b) In the abstract, the authors stated that skeletal muscle is responsible for voluntary movement. It is also responsible for non-voluntary. The abstract needs to be refocused on the mutation and on what we learn from this study. Please avoid vague statements like "we provide important insights to the pathophysiological mechanisms…" mainly when the study is descriptive and not mechanistic.

c) The author should bring up the mutation name, location and phenotype early in the introduction.

d) This reviewer also suggests that the authors refocus the introduction on the mutation location in the 3D RyR1 structure (available cryo-EM structure), if there is any nearby ligand binding site, protomers junction or any other known interacting protein partners. This will help the reader to understand how this mutation could be important for the channel's function

Reviewer 3 raised the following specific critiques:

a) The mass spec results were filtered based on very low confidence criteria: fold change 0.25 and q<0.05. which allows detection of a large number of proteins potentially irrelevant to the study. The authors should change their filtering threshold to a fold change of 0.58 (at least 50% change in the expression levels), q<0.05, and at least two unique peptides per protein.

b) In the Results section (page 6), the authors compared the protein expression profiles between EDL and soleus in wild-type mice. It is unclear why this analysis was performed. It is expected to see differences in protein expression levels when comparing a glycolytic and oxidative muscle in wild type. This is due to the properties of each muscle type.

c) The authors did not validate any of the detected changes, and whether these changes are contributing to the phenotype. Moreover, the phenotype of these mice has not been evaluated at this age. Do these mice have muscle weakness or muscle atrophy? Is there any detected protein that can be targeted by a drug?

d) The fact that RyR1 expression is decreased in all muscle types could be simply explained by the decreased muscle content for all proteins of the contractile apparatus. The histopathology of these mice needs to be evaluated for markers of atrophy, autophagy, apoptosis, and necrosis.

e) In addition to reconsidering the filtering criteria for the proteomic analysis, it is highly recommended to change the color of the heatmap to light colors to improve the readability of the figures (red and blue are very common color codes that could be used).

f) The authors could perform a GSEA analysis to have a better understanding of the enriched pathways and maybe focus on the most relevant ones. The GO pathway analysis seems to not be very informative here and the changes are subtle considering that the filter threshold was set at 0.25.

g) The discussion should be more organized and should address the pathological mechanism of the mutation and how the current study improves our understanding of the disease. For example, the section about the implication of the mutation in bone formation and remodeling is irrelevant to the current study and is an over-interpretation of the results found in skeletal muscle.

*Reviewer #1 (Recommendations for the authors):*

The subject matter is of clear scientific interest justifying full review. The objectives of the paper are clearly introduced, and the methods logically described and appropriate. The proteomic findings are striking and clear. There is a clear comparison between WT and mutant for the different muscle types. There is extensive and potentially useful reference information in the Tables and supplemental data. Subject to the detailed technical comments of the expert referee, this should be a potentially appropriate contribution for publication.

*Reviewer #2 (Recommendations for the authors):*

This study would be strengthened by inclusion of tables that summarize relative protein changes between EOM/EDL muscles and EOM/soleus muscles from WT mice as was done in Table 1 for soleus/EDL muscles from WT mice.

As there are reliable antibodies for many of the selected proteins found to be significantly increased or decreased in muscle from dHT mice, it would seem important and feasible to validate at least a subset of the proteomic findings in western blot experiments. For example, the proteome studies found JP-1 levels to be reduced in both EDL and soleus, but not in EOM; Hspa5 (BIP) levels to be increased in EDL and EOM, but unaltered in soleus; collagen levels to be reduced in EDL, but unaltered in both soleus and EOM. Such findings could readily be validated in quantitative western blot experiments.

Beyond the author-selected muscle-related processes shown in Tables 2-4, limited non-biased information is provided regarding other potential proteins and pathways altered in the different muscles evaluated from dHT mice. A more robust unbiased evaluation of all of the significantly increased/decreased proteins and the underlying pathways involved could be informative. A reactome pathway analysis comparison is shown for EDL muscles from WT and dHT mice in Supplemental Figure 1. It would be helpful if the authors could include similar reactome pathway analyses for the proteins found to be significantly increased/decreased in soleus and EOM of WT and dHT mice.

The authors provide a Venn diagram in Supplemental Figure 2 to highlight overlap between proteins that are increased and decreased across all three muscles. Interestingly, for all three muscles evaluated (EDL, soleus, EOM), more proteins were found to be increased than decreased in dHT mice (e.g. 560 proteins are increased in EOM and only 117 are decreased). In addition, while only one protein (Ryr1) was found to be decreased across all muscles, 40 proteins were found to be increased in all three muscles. It would be helpful if the authors could provide a summary table that lists these 40 proteins that are upregulated in all three muscles. Does a GO pathway analysis of these 40 proteins provide any common theme that extends across all three types muscles? Are there any common themes to come out of separate GO pathway analyses of the 848 protein differences observed in EDL, 509 protein differences observed in soleus, and 677 protein differences observed in EOM? What about the union of common proteins changes observed between EDL and EOM (144 proteins), EDL and soleus (98 proteins), and EOM and soleus (106 proteins)?

*Reviewer #3 (Recommendations for the authors):*

1) The mass spec results were filtered based on very low confidence criteria: fold change 0.25 and q<0.05. which allows detection of a large number of proteins potentially irrelevant to the study. The authors should change their filtering threshold to a fold change of 0.58 (at least 50% change in the expression levels), q<0.05, and at least two unique peptides per protein.

2) In the Results section (page 6), the authors compared the protein expression profiles between EDL and soleus in wild-type mice. It is unclear why this analysis was performed. It is expected to see differences in protein expression levels when comparing a glycolytic and oxidative muscle in wild type. This is due to the properties of each muscle type.

3) The authors did not validate any of the detected changes, and whether these changes are contributing to the phenotype. Moreover, the phenotype of these mice has not been evaluated at this age. Do these mice have muscle weakness or muscle atrophy? Is there any detected protein that can be targeted by a drug?

4) The fact that RyR1 expression is decreased in all muscle types could be simply explained by the decreased muscle content for all proteins of the contractile apparatus. The histopathology of these mice needs to be evaluated for markers of atrophy, autophagy, apoptosis, and necrosis.

5) In addition to reconsidering the filtering criteria for the proteomic analysis, it is highly recommended to change the color of the heatmap to light colors to improve the readability of the figures (red and blue are very common color codes that could be used).

6) The authors could perform a GSEA analysis to have a better understanding of the enriched pathways and maybe focus on the most relevant ones. The GO pathway analysis seems to not be very informative here and the changes are subtle considering that the filter threshold was set at 0.25.

7) The discussion should be more organized and should address the pathological mechanism of the mutation and how the current study improves our understanding of the disease. For example, the section about the implication of the mutation in bone formation and remodeling is irrelevant to the current study and is an over-interpretation of the results found in skeletal muscle.

---

## [Author Response]

Essential revisions:Reviewer (2) makes the following broad comments that may merit addressing:a) The main limitation of this study is that the results are primarily descriptive in nature, and thus, do not provide mechanistic insight into how Ryr1 disease mutations lead to the muscle-specific changes observed in the EDL, soleus and EOM proteomes.

An intrinsic feature of the high-throughput proteomic analysis technology is the generation of lists of differentially expressed proteins (DEP) in different muscles from WT and mutated mice. Although the definition of mechanistic insights related to changes of dozens of proteins is very interesting, it is a difficult task to accomplish and goes beyond the goal of the high-throughput proteomic analysis presented here. Nevertheless, the analysis of DEPs may indeed provide arguments to speculate on the pathogenesis of the phenotype linked to recessive RyR1 mutations. In the unrevised manuscript, we pointed out that the fiber type I predominance observed in congenital myopathies linked to recessive *Ryr1* mutation are consistent with the high expression level of heat shock proteins in slow twitch muscles. However, as suggested by Reviewer 3, we have removed "vague statements" from the text of the revised manuscript, concerning major insights into pathophysiological mechanisms, since we are aware that the mechanistic information, if any, that we can extract from the data set, cannot go over the intrinsic limitation of the high-throughput proteomic technology.

b) Results comparing fast twitch (EDL) and slow twitch (soleus) muscles from WT mice confirmed several known differences between the two muscle types. Similar analyses between EOM/EDL and EOM/soleus muscles from WT mice were not conducted.

We agree with the point raised by the Reviewer. In the revised manuscript we have changed figure 2. The new figure 2 shows the analysis of differentially expressed proteins in EDL, soleus and EOMs from WT mice. We have also added 2 new Tables (new supplementary 1b and 1c) and have inserted our findings in the revised Results section (page, 7, lines 156-175, pages 8 and 9).

c) While a reactome pathway analysis for proteins changes observed in EDL is shown in Supplemental Figure 1, the authors do not fully discuss the nature of the proteins and corresponding pathways impacted in the other two muscle groups analyzed.

We have now included in the revised manuscript a new figure 2 which includes the Reactome pathway analysis comparing EDL with soleus, EDL with EOM and soleus with EOM (panels C, F and I, respectively). We have also inserted into the revised manuscript a brief description of the pathways showing the greatest changes in protein content (page 7 line 156-175, pages 8 and 9). We agree that the data showing changes in protein content between the 3 muscle groups of the WT mice are important also because they validate the results of the proteomic approach. Indeed, the present results confirm that many proteins including MyHCIIb, calsequestrin 1, SERCA1, parvalbumin etc are more abundantly expressed in fast twitch EDL muscles compared to soleus. Similarly, our results confirm that EOMs are enriched in MyHC-EO as well as cardiac isoforms of ECC proteins. This point has been clarified in the revised version of the manuscript (page 8, lines 196-213; page 9 lines 214-228). Nevertheless, we would like to point out that the main focus of our study is to compare the changes of protein content induced by the presence of recessive RyR1 mutations.

Specific requests

a) This study would be strengthened by inclusion of tables that summarize relative protein changes between EOM/EDL muscles and EOM/soleus muscles from WT mice as was done in Table 1 for soleus/EDL muscles from WT mice.

In the revised version of the manuscript, we now include summary tables of relative protein changes between EDL/Soleus, EOM/EDL and EOM/Soleus (supplementary 1a, 1b and 1c).

b) As there are reliable antibodies for many of the selected proteins found to be significantly increased or decreased in muscle from dHT mice, it would seem important and feasible to validate at least a subset of the proteomic findings in western blot experiments. For example, the proteome studies found JP-1 levels to be reduced in both EDL and soleus, but not in EOM; Hspa5 (BIP) levels to be increased in EDL and EOM, but unaltered in soleus; collagen levels to be reduced in EDL, but unaltered in both soleus and EOM. Such findings could readily be validated in quantitative western blot experiments.

The unrevised manuscript and previous manuscripts (Elbaz et al. 2019; Eckhardt et al. 2020) validate the proteomic data, namely Western blot analysis with Ab against Stim1, RyR1, Casequestrin 1, DHPRa1 are consistent with the proteomic data of this study. Nevertheless, to comply with the reviewer’s request we have performed Western blot analysis to validate the differential expression of JPH1 and Col1a1 in the different muscles. As can be seen in Author response image 1, quantitative Western blot analysis with anti-JPH1 (a generous gift of Takeshima, Kyoto University, Kyoto, Japan. Ito et al., 2001 J Cell Biol 154, 1059-1067) and anti-ColIa1 (Novus Biological Catalog N° NBP2-92858) Ab confirm the data obtained by TMT mass spectrometry. Unfortunately, we did not find a reliable commercially available anti Hspa5 Ab. We tested an Ab from Novus Biological Catalog (N° NBP1-06274) but this did not recognize any band on Western blots of muscle homogenates.

**Author response image 1. sa2fig1:** Western blot analysis of muscle homogenates from WT mice probed with anti- junctophilin-1 and anti-Collagen Ia1 antibodies. Proteins in total muscle homogenates of EDL, soleus and EOM were separated on a 7.5% PAGE-SDS gel, transferred overnight onto nitrocellulose and probed with the following Ab: rabbit anti-Junctophilin-1 (1:2000, a generous gift of Takeshima, Kyoto University, Kyoto, Japan); mouse anti-collagen I alpha1 (1:2000 Novus Biological Catalog N° NBP2-92858); anti-MyHC all isoforms (1:5000; Millipore Catalog N° 41025). Western blots were incubated with the primary antibodies followed by peroxidase conjugated Protein G (Σ-Aldrich, 1:130000) or peroxidase-conjugated anti-mouse IgG (Fab Specific) Ab (Σ-Aldrich; 1:200 000). The immuno-positive bands were visualized by chemiluminescence using the WesternBright ECL HRP Substrate. For junctophilin-1 and Collagen Ia1 blots, 30 µg protein per lane were loaded; for MyHC, 5 µg protein per lane were loaded.

c) Beyond the author-selected muscle-related processes shown in Tables 2-4, limited non-biased information is provided regarding other potential proteins and pathways altered in the different muscles evaluated from dHT mice. A more robust unbiased evaluation of all of the significantly increased/decreased proteins and the underlying pathways involved could be informative. A reactome pathway analysis comparison is shown for EDL muscles from WT and dHT mice in Supplemental Figure 1. It would be helpful if the authors could include similar reactome pathway analyses for the proteins found to be significantly increased/decreased in soleus and EOM of WT and dHT mice.

As stated above in our answer to broad comments, the revised manuscript now includes the Reactome analysis comparison of EDL, EOM and Soleus muscles from WT mice. Figure 2 panels C, F and I of the revised manuscript now includes the Reactome pathway comparison between WT EDL, soleus and EOM muscles. Whereas Figure 3—figure supplement 1 of the revised manuscript shows the Reactome pathway analysis of EDL and EOM from WT and dHT mice. Since the number of proteins changing between WT and dHT soleus muscles is relatively small it does not generate a Reactome pathway. We have nevertheless analyzed in more depth differences between WT and dHT EDL, soleus and EOM applying GO analysis of proteins annotated to Biological process and GO Cellular Compartments. The new data is shown in Figure 4—figure supplement 1A and Figure 4B, C and E of the revised manuscript.

d) The authors provide a Venn diagram in Supplemental Figure 2 to highlight overlap between proteins that are increased and decreased across all three muscles. Interestingly, for all three muscles evaluated (EDL, soleus, EOM), more proteins were found to be increased than decreased in dHT mice (e.g. 560 proteins are increased in EOM and only 117 are decreased).

In Figure 4—figure supplement 1 of the revised manuscript, we now analyze GO genes annotated to biological processes of up- and downregulated proteins in each muscles type, i.e. EDL, Soleus and EOM.

In addition, while only one protein (Ryr1) was found to be decreased across all muscles, 40 proteins were found to be increased in all three muscles. It would be helpful if the authors could provide a summary table that lists these 40 proteins that are upregulated in all three muscles. Does a GO pathway analysis of these 40 proteins provide any common theme that extends across all three types muscles? Are there any common themes to come out of separate GO pathway analyses of the 848 protein differences observed in EDL, 509 protein differences observed in soleus, and 677 protein differences observed in EOM? What about the union of common proteins changes observed between EDL and EOM (144 proteins), EDL and soleus (98 proteins), and EOM and soleus (106 proteins)?

In the revised version of the manuscript, we now provide a list of the shared up-regulated proteins in the three muscle types from dHT mice (figure 4D of the revised manuscript). We also complied with the Reviewer’s request and provide GO pathways of differentially expressed protein intersects of the Venn diagram in dHT EDL, Soleus and EOM (figure 4B, figure 4C and figure 4D of the revised manuscript).

Reviewer (3) Broad comments:a) it would be useful to determine whether changes in protein levels correlated with changes in mRNA levels

We performed qPCR analysis of *Stac3* and *Cacna1s* in EDL, Soleus and EOM from WT mice (see Author response image 2). The expression of transcripts encoding *Cacna1s* and *Stac3* is approximately 9-fold higher in EDL compared to Soleus. The fold change of *Stac3* and *Cacna1s* transcripts in EDL muscles is higher compared to the differences we observed by Mass spectrometry at the protein level between EDL and Soleus. Indeed, we found that the content of the Stac3 protein in EDL is 3-fold higher compared to that in soleus. Although there is no apparent linear correlation between mRNA and protein levels, we believe that a few plausible conclusions can be drawn, namely: (i) the expression level of both transcripts and proteins is higher EDL compared to EOM and soleus muscles, respectively, (ii) the expression level of transcripts encoding *Stac3* correlate with those encoding *Cacan1s* and confirm proteomic data. In addition, the level of *Stac3* transcript does not changes between WT and dHT, confirming our proteomic data which show that Stac3 protein content in muscles from dHT is similar to that found in WT littermates. Altogether these results support the concept that the differences in Stac3 content between EDL and soleus occur at both the protein and transcript levels, namely high *Stac3* mRNA level correlates with higher protein content (EDL) and low mRNA levels correlated with low Stac3 protein content in Soleus muscles (see Author response image 2).

**Author response image 2. sa2fig2:** qPCR of *Cacna1s* and S*tac3* in muscles from WT mice. The expression levels of the transcripts encoding *Cacna1s* and *Stac3* are the highest in EDL muscles and the lowest in soleus muscles (top panels). There are no significant changes in their relative expression levels in dHT vs WT. Each symbol represents the value from of a single mouse. * p=0.028 Mann Whitney test qPCR was performed as described in Elbaz et al., 2019 (Hum Mol Genet 28, 2987-2999).

and whether or not the protein present was functional, and whether Stac3 was in fact stoichiometrically depleted in relation to Cacna1s.

We thought about this point but think that there are no plausible arguments to believe that Stac3 is not functional, one simple reason being that our WT mice do not have a phenotype which would be associated with the absence of Stac3 (Reinholt et al., PLoS One 8, e62760 2013, Nelson et al. Proc. Natl. Acad. Sci. USA 110:11881 2013).

b) In the abstract, the authors stated that skeletal muscle is responsible for voluntary movement. It is also responsible for non-voluntary. The abstract needs to be refocused on the mutation and on what we learn from this study. Please avoid vague statements like "we provide important insights to the pathophysiological mechanisms…" mainly when the study is descriptive and not mechanistic.

The abstract of the revised manuscript has been rewritten. In particular, we removed statements referring to important “pathophysiological mechanistic insight”.

c) The author should bring up the mutation name, location and phenotype early in the introduction.

In the revised manuscript we provide the information requested by the Reviewer (page 2 lines 36-38 and page 4, lines 98-102).

d) This reviewer also suggests that the authors refocus the introduction on the mutation location in the 3D RyR1 structure (available cryo-EM structure), if there is any nearby ligand binding site, protomers junction or any other known interacting protein partners. This will help the reader to understand how this mutation could be important for the channel's function

The residue Ala4329 is present inside the TMx (Auxiliary transmembrane helices) domain which spans from residue 4322 to 4370 and interposes structurally (des Georges A et al. 2016 Cell 167,145-57; Chen W, et al. 2020 EMBO Rep. 21, e49891). Although the structural resolution of the region has been improved (des Georges et al., 2016), parts of the domain still remain with no defined atomic coordinates, especially the region encompassing a.a. E4253 – F4540. Because of such undefined atomic coordinates of the region E4253-F4540, we are not able to determine the real orientation and the disposition of the amino acids in this region, including the A4329 residue. As reference, structure PDB: 5TAL of des Georges et al., 2016 was analyzed with UCSF Chimera (production version 1.16) (Pettersen et al. J. Comput. Chem. 25: 1605-1612. doi: 10.1002/jcc.20084).

Reviewer 3 raised the following specific critiques:a) The mass spec results were filtered based on very low confidence criteria: fold change 0.25 and q<0.05. which allows detection of a large number of proteins potentially irrelevant to the study. The authors should change their filtering threshold to a fold change of 0.58 (at least 50% change in the expression levels), q<0.05, and at least two unique peptides per protein.

We chose to filter our data with the FDR q<0.05, since it is the parameter most commonly adopted by the proteomic community to statistically validate changes in protein expression. The q<0.05 filter provides statistical confidence on the statistical validity of observed changes regardless of the extent of the fold change (Katrin Marcus et al. (eds.), Quantitative Methods in Proteomics, Methods in Molecular Biology, vol. 2228, https://doi.org/10.1007/978-1-0716-1024-4_1).

The reviewer proposes to set a fold change of 0.5 to detect proteins relevant to the study. We think that such a filter will miss relevant proteins. One example for all: if we set the filter for fold change of protein expression level to 0.5 we will filter out the RyR1 in the comparison between Soleus WT and Soleus from dHT mice. This result would be inconsistent with a set of data obtained by different experimental approaches including western blot analysis and qPCR (Elbaz et al. 2019) showing that the muscle phenotype of the soleus from dHT mice is in agreement with a decrease of the RyR1 protein and transcript expression level, even though it does not reach the threshold of 0.5. For this reason, we think that filtering the fold changed to 0.5 has the plausible risk of missing proteins which might be relevant to this study.

b) In the Results section (page 6), the authors compared the protein expression profiles between EDL and soleus in wild-type mice. It is unclear why this analysis was performed. It is expected to see differences in protein expression levels when comparing a glycolytic and oxidative muscle in wild type. This is due to the properties of each muscle type.

We performed the comparison of the proteome of EDL and soleus muscles, to validate the procedure of the proteomic analysis. As shows in figure 2 and Table 1 of the unrevised manuscript, our proteomic analysis indeed picked up not only the major molecular signatures of each muscle type, but also its relevance for the functional phenotype of fast and slow twitch muscles. Figure 2 and Table 1 of the unrevised manuscript confirm the molecular signatures of fast and slow twitch muscles, thus validating the functional sub-specialization of these muscle types. Therefore, we are confident of our proteomic approach results and further proceeded to analyze the proteome of EDL vs EOM, soleus vs EOM as well as the muscles from dHT mice.

c) The authors did not validate any of the detected changes, and whether these changes are contributing to the phenotype. Moreover, the phenotype of these mice has not been evaluated at this age. Do these mice have muscle weakness or muscle atrophy?

The proteomic analysis of the dHT mice is validated by published data (Elbaz et al. Hum Mol Gen 2019; Eckhardt et al. Hum Mol Gen 2020) concerning the skeletal muscle phenotype of dHT mice. In these publications, we showed that the decrease of the RyR1protein content is consistent with the decrease of force developed both in vivo and in vitro and with the smaller electrically evoked calcium transients. EDL from dHT showed a 10% decrease of wet weight, suggesting a small degree of muscle atrophy, which does not correlate with the much larger decrease of RyR1 content in the skeletal muscle homogenate. The muscle weakness is mostly explained mostly by reduction of RyR1 expression and in part by the small degree of muscle atrophy. In Elbaz et al., 2019 and Eckhardt et al., 2020 papers, mice having an age range between 8-12 weeks were characterized and in the present paper, muscles were harvested from 12 weeks old mice (page 22, line 558). See also reply to Reviewer 2, point b.

Is there any detected protein that can be targeted by a drug?

We interrogate the Drug Gene Interaction Database to analyze the differentially expressed proteins in EDL, Soleus and EOM (dHT vs WT). For example, the DGID list indicates CamKIId as potential target in EDL, Soleus and EOM. Although there are available inhibitors of this potential pharmacological target, we believe that addressing this issue goes beyond the scope of this manuscript.

d) The fact that RyR1 expression is decreased in all muscle types could be simply explained by the decreased muscle content for all proteins of the contractile apparatus. The histopathology of these mice needs to be evaluated for markers of atrophy, autophagy, apoptosis, and necrosis.

The proteomic analysis of the dHT mice is validated by data (Elbaz et al. Hum Mol Gen 2019; Eckhardt et al. Hum Mol Gen 2020) concerning the skeletal muscle phenotype of the dHT mice. Elbaz et al. (2019) showed that in EDL and Soleus muscle from dHT mice there are ultrastructural changes referable to core-like structures, but no major signatures of atrophy and or necrosis (Author response image 3). We would like to point out that Reactome pathway analysis and GO analysis did not identify terms annotated to either autophagy, apoptosis or necrosis.

**Author response image 3. sa2fig3:** Histological staining of EDL and soleus muscles from EDL and soleus. Cross sectional area of HandE stained muscle sections from WT and dHT mice. No evidence of inflammation or necrosis is present in muscles from dHT mice. Bar = 100 µm. Muscles were isolated and embedded in OCT and deep-frozen in 2-methylbutane. Transver 10 µm thick muscle sections were made with a Leica Cryostat (CM1950).

e) In addition to reconsidering the filtering criteria for the proteomic analysis, it is highly recommended to change the color of the heatmap to light colors to improve the readability of the figures (red and blue are very common color codes that could be used).

We used yellow blue for the heat maps because such colors are more compatible for people with impaired color vision (tritanomaly and tritanopia).

f) The authors could perform a GSEA analysis to have a better understanding of the enriched pathways and maybe focus on the most relevant ones. The GO pathway analysis seems to not be very informative here and the changes are subtle considering that the filter threshold was set at 0.25.

The reviewer suggests to use the GSEA tool to explore enrichment pathway since GO pathway was not informative. We believe that the use of the GSEA tool and other tools will face the same problem, i.e. the enrichment terms are not informative for EC coupling and other muscle physiology terms because most of the annotated terms in these tools are related to cancer or immunological pathways regardless of the filter threshold relative to fold changes. We did use GSEA tool and we did not pick up significant terms relating to EC coupling and neuromuscular disorders linked to RyR1 mutations

g) The discussion should be more organized and should address the pathological mechanism of the mutation and how the current study improves our understanding of the disease. For example, the section about the implication of the mutation in bone formation and remodeling is irrelevant to the current study and is an over-interpretation of the results found in skeletal muscle.

The discussion of the revised manuscript has been reorganized with new paragraphs. In addition, in the revised manuscript we removed the sentence referring to the effects of RyR1 mutations on bone formation and remodeling.